# On Tighter Generalization Bounds for Deep Neural Networks: CNNs, ResNets, and Beyond

## Abstract

We propose a generalization error bound for a general family of deep neural networks based on the depth and width of the networks, as well as the spectral norm of weight matrices. Through introducing a novel characterization of the Lipschitz properties of neural network family, we achieve a tighter generalization error bound. We further obtain a result that is free of linear dependence on norms for bounded losses. Besides the general deep neural networks, our results can be applied to derive new bounds for several popular architectures, including convolutional neural networks (CNNs), residual networks (ResNets), and hyperspherical networks (SphereNets). When achieving same generalization errors with previous arts, our bounds allow for the choice of much larger parameter spaces of weight matrices, inducing potentially stronger expressive ability for neural networks.

## 1 Introduction

We aim to provide a theoretical justification for the enormous success of deep neural networks (DNNs) in real world applications (He et al., 2016; Collobert et al., 2011; Goodfellow et al., 2016). In particular, our paper focuses on the generalization performance of a general class of DNNs. The generalization bound is a powerful tool to characterize the predictive performance of a class of learning models for unseen data. Early studies investigate the generalization ability of shallow neural networks with no more than one hidden layer (Bartlett, 1998; Anthony & Bartlett, 2009). More recently, studies on the generalization bounds of deep neural networks have received increasing attention (Dinh et al., 2017; Bartlett et al., 2017; Golowich et al., 2017; Neyshabur et al., 2015; 2017). There are two major questions of our interest in these analysis of the generalization bounds:

(Q1) *Can we establish tighter generalization error bounds for deep neural networks in terms of the network dimensions and structure of the weight matrices?*

(Q2) *Can we develop generalization bounds for neural networks with special architectures?*

For (Q1), (Neyshabur et al., 2015; Bartlett et al., 2017; Neyshabur et al., 2017; Golowich et al., 2017) have established results that characterize the generalization bounds in terms of the depth $D$ and width $p$ of networks and norms of rank-$r$ weight matrices. For example, Neyshabur et al. (2015) provide an exponential bound on $D$ based on the Frobenius norm $\|W_d\|_F$, where $W_d$ is the weight matrix of $d$-th layer; Bartlett et al. (2017); Neyshabur et al. (2017) provide a polynomial bound on $p$ and $D$ based on $\|W_d\|_2$ (spectral norm) and $\|W_d\|_{2,1}$ (sum of the Euclidean norms for all rows of $W_d$). Golowich et al. (2017) provide a nearly size independent bound based on $\|W_d\|_F$. Nevertheless, the generalization bound depends on other than the spectral norms of the weight matrices may be too loose. In specific, $\|W_d\|_F$ ($\|W_d\|_{2,1}$) is in general $\sqrt{r}$ ($r$) times larger than $\|W_d\|_2$. Given $m$ training data points and suppose $\|W_d\|_2 = 1$ for ease of discussion, Bartlett et al. (2017) and Neyshabur et al. (2017) demonstrate generalization error bounds as $\widetilde{\mathcal{O}}(\sqrt{D^3 pr/m})$, and Golowich et al. (2017) achieve a bound $\widetilde{\mathcal{O}}(r^{D/2} \min(m^{-1/4}, \sqrt{D/m}))$, where $\widetilde{\mathcal{O}}(\cdot)$ represents the rate by ignoring logarithmic factors. In comparison, we show a tighter generalization error bound as $\widetilde{\mathcal{O}}(\sqrt{Dpr/m})$, which is significantly smaller than existing results and achieved based on a new Lipschitz analysis for DNNs in terms of both the input and weight matrices. We notice that some recent results characterize the generalization bound in more structured ways, e.g., by considering specific error-resilience parameters (Arora et al., 2018), which can achieve empirically improved

Table 1: Comparison of existing results with ours on norm based generalization error bounds for DNNs. For ease of illustration, we suppose the upper bound of input norm $R$ and the Lipschitz constant $\frac{1}{\gamma}$ of the class of loss functions $g_\gamma$ are generic constants. We use $B_{d,2}$, $B_{d,F}$, and $B_{d,2\to1}$ as the upper bounds of $\|W_d\|_2$, $\|W_d\|_F$, and $\|W_d\|_{2,1}$ respectively. For notational convenience, we suppose the width $p_d = p$ for all layers $d = 1, \ldots, D$. We further show the results when $\|W_d\|_2 = 1$ for all $d = 1, \ldots, D$, where $\|W_d\|_F = \Theta(\sqrt{r})$ and $\|W_d\|_{2,1} = \Theta(r)$ in generic scenarios.

| Generalization Bound | Original Results | $\|W_d\|_2 = 1$ |
|---|---|---|
| Neyshabur et al. (2015) | $\mathcal{O}\left(\frac{2^D \cdot \Pi_{d=1}^D B_{d,F}}{\sqrt{m}}\right)$ | $\mathcal{O}\left(\frac{2^D \cdot r^{D/2}}{\sqrt{m}}\right)$ |
| Bartlett et al. (2017) | $\widetilde{\mathcal{O}}\left(\frac{\Pi_{d=1}^D B_{d,2}}{\sqrt{m}}\left(\sum_{d=1}^D \frac{B_{d,2\to1}^{2/3}}{B_{d,2}^{2/3}}\right)^{3/2}\right)$ | $\widetilde{\mathcal{O}}\left(\frac{\sqrt{D^3 pr}}{\sqrt{m}}\right)$ |
| Neyshabur et al. (2017) | $\widetilde{\mathcal{O}}\left(\frac{\Pi_{d=1}^D B_{d,2}}{\sqrt{m}}\sqrt{D^2 p \sum_{d=1}^D \frac{B_{d,F}^2}{B_{d,2}^2}}\right)$ | $\widetilde{\mathcal{O}}\left(\frac{\sqrt{D^3 pr}}{\sqrt{m}}\right)$ |
| Golowich et al. (2017) | $\widetilde{\mathcal{O}}\left(\Pi_{d=1}^D B_{d,F} \cdot \min\left\{\frac{1}{\sqrt[4]{m}}, \sqrt{\frac{D}{m}}\right\}\right)$ | $\widetilde{\mathcal{O}}\left(r^{D/2} \cdot \min\left\{\frac{1}{\sqrt[4]{m}}, \sqrt{\frac{D}{m}}\right\}\right)$ |
| Our results | Theorem 1: $\widetilde{\mathcal{O}}\left(\frac{\Pi_{d=1}^D B_{d,2}\sqrt{Dpr}}{\sqrt{m}}\right)$ | $\widetilde{\mathcal{O}}\left(\frac{\sqrt{Dpr}}{\sqrt{m}}\right)$ |

generalization bounds than existing ones based on the norms of weight matrices. However, it is not clear how the weight matrices explicitly control these parameters, which makes the results less interpretable. Thus, we do not compare with these types of results. We summarize the comparison between existing norm based generalization bounds with our results in Table 1, as well as the results when $\|W_d\|_2 = 1$ for more explicit comparison in terms of the network sizes (i.e, depth and width).

For (Q2), we consider several widely used architectures to demonstrate, including convolutional neural networks (CNNs) (Krizhevsky et al., 2012), residual networks (ResNets) (He et al., 2016), and hyperspherical networks (SphereNets) (Liu et al., 2017b). By taking their structures of weight matrices into consideration, we provide tight characterization of their resulting capacities. In particular, we consider orthogonal filters and normalized weight matrices, which show good performance in both optimization and generalization (Mishkin & Matas, 2015; Xie et al., 2017). This is closely related with normalization frameworks, e.g., batch normalization (Ioffe & Szegedy, 2015) and layer normalization (Ba et al., 2016), which have achieved great empirical performance (Liu et al., 2017a; He et al., 2016). Take CNNs as an example. By incorporating the orthogonal structure of convolutional filters, we achieve $\widetilde{\mathcal{O}}((\frac{k}{s})^{\frac{D}{2}}\sqrt{Dk^2}/\sqrt{m})$, while Bartlett et al. (2017); Neyshabur et al. (2017) achieve $\widetilde{\mathcal{O}}((\frac{k}{s})^{\frac{D-1}{2}}\sqrt{D^3 p^2}/\sqrt{m})$ and Golowich et al. (2017) achieve $\widetilde{\mathcal{O}}\left(p^{\frac{D}{2}}\min\left\{\frac{1}{\sqrt[4]{m}}, \sqrt{\frac{D}{m}}\right\}\right)$ (rank$(W_d) = p$ in CNNs), where $k$ is the filter size that satisfies $k \ll p$ and $s$ is stride size that is usually of the same order with $k$; see Section 4.1 for details. Here we achieve stronger results in terms of both depth $D$ and width $p$ for CNNs, where our bound only depend on $k$ rather than $p$. Some recent result achieved results that is free of the linear dependence on the weight matrix norms by considering networks with bounded outputs (Zhou & Feng, 2018). We can achieve similar results using bounded loss functions as discussed in Section 3.2, but do not restrict ourselves to this scenario in general. Analogous improvement is also attained for ResNets and SphereNets. In addition, we consider some widely used operations for width expansion and reduction, e.g., padding and pooling, and show that they do not increase the generalization bound. Further numerical evaluation is provided for quantitative comparison in Section 4.5.

Our tighter bounds result in potentially stronger expressive power, hence higher training/testing accuracy for the DNNs. In particular, when achieving the same order of generalization errors, we allow the choice of a larger parameter space with deeper/wider networks and larger matrix spectral norms. We further show numerically that a larger parameter space can lead to better empirical performance. Quantitative analysis for the expressive power of DNNs is of great interest on its own, which includes (but not limited to) studying how well DNNs can approximate general class of functions and distributions (Cybenko, 1989; Hornik et al., 1989; Funahashi, 1989; Barron, 1993; 1994; Lee et al., 2017; Petersen & Voigtlaender, 2017; Hanin & Sellke, 2017), and quantifying the computation hardness of learning neural networks; see e.g., Shamir (2016); Eldan & Shamir (2016); Song et al. (2017). We defer our investigation toward this to future efforts.

**Notation.** Given an integer $n > 0$, we define $[n] = \{1, \ldots, n\}$. Given a matrix $A \in \mathbb{R}^{n \times m}$, we denote $\|A\|$ as a generic norm, $\|A\|_2$ as the spectral norm, $\|A\|_{\mathrm{F}}$ as the Frobenius norm, and $\|A\|_{2,1} = \sum_{i=1}^{n} \|A_{i*}\|_2$. We use the standard notations $\mathcal{O}(\cdot)$, $\Theta(\cdot)$, and $\Omega(\cdot)$ to denote limiting behaviors ignoring constants, and $\widetilde{\mathcal{O}}(\cdot)$, $\widetilde{\Theta}(\cdot)$ and $\widetilde{\Omega}(\cdot)$ to further ignore logarithmic factors.

## 2 PRELIMINARIES

We provide a brief description of the DNNs. Given an input $x \in \mathbb{R}^{p_0}$, the output of a $D$-layer network is defined as $f_D f(\mathcal{W}_D, x) = f_{W_D}(\cdots f_{W_1}(x)) \in \mathbb{R}^{p_D}$, where $f_{W_d}(y) = \sigma_d(W_d \cdot y) : \mathbb{R}^{p_{d-1}} \to \mathbb{R}^{p_d}$ with an entry-wise activation function $\sigma_d(\cdot)$. We specify $\sigma_d$ as the rectified linear unit (ReLU) activation (Nair & Hinton, 2010) for ease of discussion. The extension to more general activations, e.g., Lipschitz continuous functions, is straightforward. Then we denote DNNs with bounded weight matrices $\mathcal{W}_D = \{W_d \in \mathbb{R}^{p_d \times p_{d-1}}\}_{d=1}^{D}$ and ranks as

$$\mathcal{F}_{D, \|\cdot\|} = \{f(\mathcal{W}_D, x) \mid \forall d \in [D], W_d \in \mathcal{W}_D, \|W_d\| \leq B_d, \mathrm{rank}(W_d) \leq r_d\}, \quad (1)$$

where $x \in \mathbb{R}^{p_0}$ is an input, and $\{B_d\}$ are real positive constants. We will specify the norm $\|\cdot\|$ and the corresponding upper bounds $B_d$, e.g., $\|\cdot\|_2$ and $B_{d,2}$, or $\|\cdot\|_{\mathrm{F}}$ and $B_{d,\mathrm{F}}$, when necessary.

Given a loss function $g(\cdot, \cdot)$, we denote a class of loss functions measuring the discrepancy between a DNN's output $f(\mathcal{W}_D, x)$ and the corresponding observation $y \in \mathcal{Y}_m$ for a given input $x \in \mathcal{X}_m$ as

$$\mathcal{G}(\mathcal{F}_{D, \|\cdot\|}) = \{g(f(\mathcal{W}_D, x), y) \in \mathbb{R} \mid x \in \mathcal{X}_m, y \in \mathcal{Y}_m, f(\cdot, \cdot) \in \mathcal{F}_{D, \|\cdot\|}\},$$

where the sets of bounded inputs $\mathcal{X}_m$ and the corresponding observations $\mathcal{Y}_m$ are

$$\mathcal{X}_m = \{x_i \in \mathbb{R}^{p_0} \mid \|x_i\|_2 \leq R \text{ for all } i \in [m]\} \subset \mathcal{X} \text{ and } \mathcal{Y}_m = \{y_i \in [p_D] \text{ for all } i \in [m]\} \subset \mathcal{Y}.$$

Then the empirical Rademacher complexity (ERC) of $\mathcal{G}(\mathcal{F}_{D, \|\cdot\|})$ given $\mathcal{X}_m$ and $\mathcal{Y}_m$ is

$$\mathcal{R}_m(\mathcal{G}(\mathcal{F}_{D, \|\cdot\|})) = \mathop{\mathbb{E}}_{\epsilon \in \{\pm 1\}^m} \left[ \sup_{f(\cdot, \cdot) \in \mathcal{F}_{D, \|\cdot\|}} \left| \frac{1}{m} \sum_{i=1}^{m} \epsilon_i \cdot g(f(\mathcal{W}_D, x_i), y_i) \right| \, \bigg| \, \mathcal{X}_m, \mathcal{Y}_m \right], \quad (2)$$

where $\{\pm 1\}^m \in \mathbb{R}^m$ is the set of vectors only containing entries $+1$ and $-1$, and $\epsilon \in \mathbb{R}^m$ is a vector with Rademacher entries, i.e., $\epsilon_i = +1$ or $-1$ with equal probabilities.

Take the classification as an example. For multi-class classification, suppose $p_D = N_{\mathrm{class}}$ is the number of classes. Consider $g$ with bounded outputs, namely the *ramp risk*. Specifically, for an input $x$ belonging to class $y \in [N_{\mathrm{class}}]$, we denote $\nu = (f(\mathcal{W}_D, x))_y - \max_{i \neq y} (f(\mathcal{W}_D, x))_i$. For a given real value $\gamma > 0$, the class of ramp risk functions with parameter $\gamma$ is $\mathcal{G}_\gamma(\mathcal{F}_{D, \|\cdot\|}) = \{g_\gamma(f(\mathcal{W}_D, x), y) \mid f_D \in \mathcal{F}_{D, \|\cdot\|}\}$, where $g_\gamma$ is $\frac{1}{\gamma}$-Lipschitz continuous, defined in (3). For conve-

$$g_\gamma(f(\mathcal{W}_D, x), y) = \begin{cases} 0, & \nu > \gamma \\ 1 - \frac{\nu}{\gamma}, & \nu \in [0, \gamma] \\ 1, & \nu < 0, \end{cases} \quad (3)$$

nience, we denote $g_\gamma(f(\mathcal{W}_D, x), y)$ as $g_\gamma(f(\mathcal{W}_D, x))$ (or $g_\gamma$) in the rest of the paper.

Then the generalization error bound for PAC learning (Bartlett et al., 2017) (Lemma 3.1) states the following. Given any real $\delta \in (0, 1)$ and $g_\gamma$, with probability at least $1 - \delta$, we have that for any $f(\cdot, \cdot) \in \mathcal{F}_{D, \|\cdot\|}$, the generalization error is upper bounded with respect to (w.r.t.) the ERC satisfies

$$\mathbb{E}[g_\gamma(f(\mathcal{W}_D, x))] - \frac{1}{m} \sum_{i=1}^{m} g_\gamma(f(\mathcal{W}_D, x_i)) \leq 2\mathcal{R}_m(\mathcal{G}_\gamma(\mathcal{F}_{D, \|\cdot\|})) + 3\sqrt{\frac{\log(\frac{2}{\delta})}{2m}}. \quad (4)$$

The right hand side of (4) is viewed as a guaranteed error bound for the gap between the testing and the empirical training performance. Since the ERC is generally the dominating term in (4), a small $\mathcal{R}_m$ is desired for DNNs given the loss function $g_\gamma$. Analogous results hold for regression tasks; see e.g., Kearns & Vazirani (1994); Mohri et al. (2012) for details.

## 3 GENERALIZATION ERROR BOUND FOR DNNS

We introduce some additional notations first. Given any two layers $i, j \in [D]$ and input $x$, we denote $J_{i:j}^x$ as the Jacobian from layer $i$ to layer $j$, i.e., $f_{W_j}(\cdots f_{W_i}(x)) = J_{i:j}^x \cdot x$. For convenience, we denote $f_{W_i}(x) = J_{i,i}^x \cdot x$ when $i = j$ and denote $J_{i:j}^x = I$ when $i > j$. Next, we denote $B_{i:j,2}^{\mathrm{Jac}, x}$ as an upper bound of the norm of Jacobian for input $x$ over the parameter, i.e., $\sup_{\mathcal{W}_D} \|J_{i,i}^x\|_2 \leq B_{i:j,2}^{\mathrm{Jac}, x}$.

### 3.1 A TIGHTER ERC BOUND FOR DNNS

We first provide the ERC bound for the class of DNNs defined in (1) and Lipschitz loss functions in the following theorem. The proof is provided in Appendix B.

**Theorem 1.** Let $g_\gamma$ be a $\frac{1}{\gamma}$-Lipschitz loss function and $\mathcal{F}_{D,\|\cdot\|_2}$ be the class of DNNs defined in (1), $p_d = p$, $r_d = r$ for all $d \in [D]$, $B_{\backslash d,2}^{\text{Jac}} = \max_{d \in [D], x \in \mathcal{X}_m} B_{1:(d-1),2}^{\text{Jac},x} B_{(d+1):D,2}^{\text{Jac},x}$, and $C^{\text{Net}} = \frac{B_{\backslash d,2}^{\text{Jac}} \cdot R\sqrt{Dm/r} \cdot \max_d B_{d,2}}{\sup_{f \in \mathcal{F}_{D,\|\cdot\|_2}, x \in \mathcal{X}_m} g_\gamma(f(\mathcal{W}_D, x))}$. Then the ERC satisfies

$$\mathcal{R}_m\left(\mathcal{G}_\gamma\left(\mathcal{F}_{D,\|\cdot\|_2}\right)\right) = \mathcal{O}\left(\frac{R \cdot \prod_{d=1}^D B_{d,2}\sqrt{Dpr \log C^{\text{Net}}}}{\gamma\sqrt{m}}\right). \tag{5}$$

**Remark 1.** Note that $C^{\text{Net}}$ depends on the norm of Jacobian, which is significantly smaller than the product of matrix norms that is exponential on $D$ in general. For example, when we obtain the network from stochastic gradient descent using randomly initialized weights, then $B_{\backslash d,2}^{\text{Jac}} \ll \prod_d B_{d,2}$. Empirical distributions of $B_{\backslash d,2}^{\text{Jac}}$ and $\prod_d B_{d,2}$ are provided in Appendix A.2, where $B_{\backslash d,2}^{\text{Jac}}$'s are constants that are orders of magnitude smaller than $\prod_d B_{d,2}$. Further experiment in Appendix A.3 shows that $B_{\backslash d,2}^{\text{Jac}}$ has a dependence slower than some low degree poly(depth), rather than exponential on the depth as in $\prod_d B_{d,2}$. Thus, $\log C^{\text{Net}}$ can be considered as a constant almost independent of $D$ in practice. Even in the worst case that $B_{\backslash d,2}^{\text{Jac}} \approx \prod_d B_{d,2}$ (this almost never happens in practice), our bound is still tighter than existing spectral norm based bounds (Bartlett et al., 2017; Neyshabur et al., 2017) by an order of $\sqrt{D}$. Also note that $C^{\text{Net}}$ is a quantity (including $B_{\backslash d,2}^{\text{Jac}}$) only depending on the training dataset, which is due to the fact that the ERC only depends on the training dataset.

For convenience, we treat $R/\gamma$ as a constant here. We achieve $\widetilde{\mathcal{O}}(\Pi_{d=1}^D B_{d,2} \cdot \sqrt{Dpr/m})$ in Theorem 1, which is tighter than existing results based on the network sizes and norms of weight matrices, as shown in Table 1. In particular, Neyshabur et al. (2015) show an exponential dependence on $D$, i.e., $\mathcal{O}(2^D \Pi_{d=1}^D B_{d,\text{F}}/\sqrt{m})$, which can be significantly larger than ours. Bartlett et al. (2017); Neyshabur et al. (2017) demonstrate polynomial dependence on sizes and the spectral norm of weights, i.e., $\widetilde{\mathcal{O}}(\Pi_{d=1}^D B_{d,2} \cdot \sqrt{D^3 pr/m})$. Our result in (5) is tighter by an order of $D$, which is significant in practice. More recently, Golowich et al. (2017) demonstrate a bound w.r.t the Frobenius norm as $\widetilde{\mathcal{O}}(\Pi_{d=1}^D B_{d,\text{F}} \cdot \min\{\sqrt{\frac{D}{m}}, m^{-\frac{1}{4}} \cdot \log^{\frac{3}{4}}(m)\sqrt{\log(C)}\})$, where $C = \frac{R \cdot \Pi_{d=1}^D B_{d,\text{F}}}{\sup_{x \in \mathcal{X}_m} \|f(\mathcal{W}_D, x)\|_2}$. This has a tighter dependence on network sizes. Nevertheless, $\|W_d\|_\text{F}$ is generally $\sqrt{r}$ times larger than $\|W_d\|_2$, which results in an exponential dependence $p^{D/2}$ compared with the bound based on the spectral norm. Moreover, $\log(C)$ is linear on $D$ except that the stable ranks $\|W_d\|_\text{F}/\|W_d\|_2$ across all layers are close to 1 (rather than almost independent on $D$ as in (5) without low-rank constraints). In addition, it has $m^{-\frac{1}{4}}$ dependence rather than $m^{-\frac{1}{2}}$ except when $D = \mathcal{O}(\sqrt{m})$. Note that our bound is based on a novel characterization of Lipschitz properties of DNNs, which may be of independent interest from the learning theory point of view. We refer to Appendix B for details.

We also remark that when achieving the same order of generalization errors, we allow the choices of larger dimensions $(D, p)$ and spectral norms of weight matrices, which lead to stronger expressive power for DNNs. For example, when achieving the same bound with $\|W_d\|_2 = 1$ in spectral norm based results (e.g. in ours) and $\|W_d\|_\text{F} = 1$ in Frobenius norm based results (e.g., in Golowich et al. (2017)), they only have $\|W_d\|_2 = \mathcal{O}(1/\sqrt{r})$ in Frobenius norm based results. The later results in a much smaller space for eligible weight matrices as $r$ is of order $p$ in general (i.e., $r = \delta p$ for some constant $\delta \in (0,1)$), which may lead to weaker expressive ability of DNNs. We also demonstrate numerically in Section 4.5 that when norms of weight matrices are constrained to be very small, both training and testing performance degrade significantly. A quantitative analysis for the tradeoff between the expressive ability and the generalization for DNNs is deferred to a future effort.

### 3.2 A SPECTRAL NORM FREE ERC BOUND

When, in addition, the loss function is bounded, we have the ERC bound free of the linear dependence on the spectral norm, as in the following corollary. The proof is provided in Appendix C.

**Corollary 1.** In addition to the conditions in Theorem 1, suppose we further let $g_\gamma$ be bounded, i.e., $|g_\gamma| \leq b$. Then the ERC satisfies

$$\mathcal{R}_m\left(\mathcal{G}_\gamma\left(\mathcal{F}_{D,\|\cdot\|_2}\right)\right) = \mathcal{O}\left(\min\left\{\frac{R\prod_{d=1}^D B_{d,2}}{\gamma}, b\right\} \cdot \sqrt{\frac{Dpr \log C^{\text{Net}}}{m}}\right). \tag{6}$$

The boundedness of $\mathcal{G}_\gamma$ holds for certain loss functions, e.g., the ramp risk defined in (3). When $b$ is constant (e.g., $b = 1$ for the ramp risk) and $R \prod_{d=1}^D B_{d,2} > \gamma$, we have that the ERC reduces to $\widetilde{\mathcal{O}}(\sqrt{Dpr/m})$. This is close to the VC dimension of DNNs, which can be significantly tighter than existing norm based bounds in general. Similar norm free results hold for the architectures discussed in Section 4 using argument for Corollary 1, which we skip due to space limit. Moreover, our bound (6) is also tighter than recent results that are free of linear dependence on $\prod_{d=1}^D B_{d,2}$ (Zhou & Feng, 2018; Arora et al., 2018). Specifically, Zhou & Feng (2018) show that the generalization bound for CNNs is $\widetilde{\mathcal{O}}(D\sqrt{pr^2/m})$, which results in a bound larger than (6) by $\mathcal{O}(\sqrt{Dr})$. Arora et al. (2018) derive a bound for a compressed network in terms of some error-resilience parameters, which is $\widetilde{\mathcal{O}}(\sqrt{D^3 p^2/m})$ since the cushion parameter therein is of the order $\mu = \mathcal{O}(1/\sqrt{p})$. Further discussion is provided in Appendix A.1.

## 4 EXPLORING NETWORK STRUCTURES

The generic result in Section 3 does not highlight explicitly the potential impacts for specific structures of the networks. In this section, we consider several popular architectures of DNNs, including convolutional neural networks (CNNs) (Krizhevsky et al., 2012), residual networks (ResNets) (He et al., 2016), and hyperspherical networks (SphereNets) (Liu et al., 2017b), and provide sharp characterization of the corresponding generalization bounds. In particular, we consider orthogonal filters and normalized weight matrices, which have shown good performance in both optimization and generalization (Mishkin & Matas, 2015; Huang et al., 2017). Such constraints can be enforced using regularizations on filters and weight matrices, which is very efficient to implement in practice. This is also closely related with normalization approaches, e.g., batch normalization (Ioffe & Szegedy, 2015) and layer normalization (Ba et al., 2016), which have achieved tremendous empirical success.

### 4.1 CNNs WITH ORTHOGONAL FILTERS

CNNs are one of the most powerful architectures in deep learning, especially in tasks related to images and videos (Goodfellow et al., 2016). We consider a tight characterization of the generalization bound for CNNs by generating the weight matrices using unit norm orthogonal filters, which has shown great empirical performance (Huang et al., 2017; Xie et al., 2017). Specifically, we generate the weight matrices using a circulant approach, as follows. For the convolutional operation at the $d$-th layer, we have $n_d$ channels of convolution filters, each of which is generated from a $k_d$-dimensional feature using a stride side $s_d$. Suppose that $s_d$ divides both $k_d$ and $p_{d-1}$, i.e., $\frac{k_{d-1}}{s_d}$ and $\frac{p_{d-1}}{s_d}$ are integers, then we have $p_d = \frac{n_d \cdot p_{d-1}}{s_d}$. This is equivalent to fixing the weight matrix at the $d$-th layer to be generated as in (7), where for all $j \in [n_d]$, each $W_d^{(j)} \in \mathbb{R}^{\frac{p_{d-1}}{s_d} \times p_{d-1}}$ is formed in a circulant-like way using a vector $w^{(d,j)} \in \mathbb{R}^{k_d}$ with unit norms for all $j$ as in (8).

$$W_d = \left[ W_d^{(1)\top} \cdots W_d^{(n_d)\top} \right]^\top \in \mathbb{R}^{p_d \times p_{d-1}}, \quad (7)$$

$$W_d^{(j)} = \begin{bmatrix} w^{(d,j)} \underbrace{0 \cdots \cdots \cdots \cdots \cdots 0}_{\in \mathbb{R}^{p_{d-1}-k_d}} \\ \underbrace{0 \cdots 0}_{\in \mathbb{R}^{s_d}} w^{(d,j)} \underbrace{0 \cdots \cdots \cdots \cdots 0}_{\in \mathbb{R}^{p_{d-1}-k_d-s_d}} \\ \vdots \\ w^{(d,j)}_{(s_d+1):k_d} \underbrace{0 \cdots \cdots \cdots \cdots 0}_{\in \mathbb{R}^{p_{d-1}-k_d}} w^{(d,j)}_{1:s_d} \end{bmatrix}. \quad (8)$$

When the stride size $s_d = 1$, $W_d^{(j)}$ corresponds to a standard circulant matrix (Davis, 2012). The following lemma establishes that when $\left\{ w^{(d,j)} \right\}_{j=1}^{n_d}$ are orthogonal vectors with unit Euclidean norms, the generalization bound only depend on $s_d$ and $k_d$ that are independent of the width $p_d$. The proof is provided in Appendix D.

**Corollary 2.** Let $g_\gamma$ be a $\frac{1}{\gamma}$-Lipschitz and bounded loss function, i.e., $|g_\gamma| \le b$, and $\mathcal{F}_{D,\|\cdot\|_2}$ be the class of CNNs defined in (1). Suppose the weight matrices in CNNs are formed as in (7) and (8) with $s_d = s$, $k_d = k$, and $s$ divides both $k$ and $p_d$ for all $d \in [D]$, where $\left\{ w^{(d,j)} \right\}_{j=1}^{n_d}$ satisfies $w^{(j)\top} w^{(i)} = 0$ for all $i,j \in [n_d]$ and $i \ne j$ with $\left\| w^{(d,j)} \right\|_2 = 1$ for all $j \le n_d$. Denote $C^{\text{Net}} = \frac{B_{\setminus d,2}^{\text{Jac}} \cdot R\sqrt{Dm/s}}{\sup_{f \in \mathcal{F}_{D,\|\cdot\|_2}, x \in \mathcal{X}_m} g_\gamma(f(\mathcal{W}_D, x))}$. Then the ERC for CNNs satisfies

$$\mathcal{R}_m \left( \mathcal{G}_\gamma \left( \mathcal{F}_{D,\|\cdot\|_2} \right) \right) = \mathcal{O} \left( \min \left\{ \frac{R(k/s)^{D/2}}{\gamma}, b \right\} \cdot \sqrt{\frac{k \sum_{d=1}^D n_d \cdot \log C^{\text{Net}}}{m}} \right).$$

Since $n_d \leq k$ in our setting, the ERC for CNNs is proportional to $\sqrt{Dk^2}$ instead of $\sqrt{Dpr}$. For the orthogonal filtered considered in Corollary 2, we have $\|W_d\|_F = \sqrt{p_d}$ and $\|W_d\|_{2,1} = p_d$, which lead to the bounds of CNNs in existing results in Table 2. In practice, one usually has $k_d \ll p_d$, which exhibit a significant improvement over existing results, i.e., $\sqrt{Dk^2} \ll \sqrt{D^3 p^2}$. Even without the orthogonal constraint on filters, the rank $r$ in CNNs is usually of the same order with width $p$, which also makes the existing bound undesirable. On the other hand, it is widely used in practice that $k_d = \mu s_d$ for some small constant $\mu \geq 1$ in CNNs, then we have $(k_d/s_d)^{D/2} \ll p^{D/2}$ resulted from $\mathcal{R}_m \left( \mathcal{G}_\gamma \left( \mathcal{F}_{D, \|\cdot\|_F} \right) \right)$.

**Remark 2.** We consider vector input in Corollary 2. For matrix inputs, e.g., images, similar results hold by considering vectorized input and permuting columns of $W_d$. Specifically, suppose $\sqrt{k_d}$ and $\sqrt{p_{d-1}}$ are integers for ease of discussion. Consider the input as a $p_{d-1}$ dimensional vector obtained by vectorizing a $\sqrt{p_{d-1}} \times \sqrt{p_{d-1}}$ input matrix. When the 2-dimensional (matrix) convolutional filters are of size $\sqrt{k_d} \times \sqrt{k_d}$, we form the rows of each $W_d^{(j)}$ by concatenating $\sqrt{k_d}$ vectors $\{w^{(j,i)}\}_{i=1}^{\sqrt{k_d}}$ padded with 0's, each of which is a concatenation of one row of the filter of size $\sqrt{k_d}$ with some zeros as follows:

Table 2: Comparison with existing norm based bounds of CNNs. We suppose $R$ and $\gamma$ are generic constants for ease of illustration. The results of CNNs in existing works are obtained by substituting the corresponding norms of the weight matrices generated by orthogonal filters, i.e., $\|W_d\|_2 = \sqrt{k/s}$, $\|W_d\|_F = \sqrt{p}$, and $\|W_d\|_{2,1} = p$.

| Generalization Bound | CNNs |
|---|---|
| Neyshabur et al. (2015) | $\mathcal{O}\left( \frac{2^D \cdot p^{\frac{D}{2}}}{\sqrt{m}} \right)$ |
| Bartlett et al. (2017) | $\widetilde{\mathcal{O}}\left( \frac{\left(\frac{k}{s}\right)^{\frac{D-1}{2}} \cdot \sqrt{D^3 p^2}}{\sqrt{m}} \right)$ |
| Neyshabur et al. (2017) | $\widetilde{\mathcal{O}}\left( \frac{\left(\frac{k}{s}\right)^{\frac{D-1}{2}} \cdot \sqrt{D^3 p^2}}{\sqrt{m}} \right)$ |
| Golowich et al. (2017) | $\widetilde{\mathcal{O}}\left( p^{\frac{D}{2}} \min\left\{ \frac{1}{\sqrt[4]{m}}, \sqrt{\frac{D}{m}} \right\} \right)$ |
| Our results | $\widetilde{\mathcal{O}}\left( \frac{\left(\frac{k}{s}\right)^{\frac{D}{2}} \sqrt{Dk^2}}{\sqrt{m}} \right)$ |

$$\underbrace{w^{(j,1)}}_{\in \mathbb{R}^{\sqrt{k_d}}} \underbrace{0 \cdots \cdots 0}_{\in \mathbb{R}^{\sqrt{\frac{p_{d-1}}{k_d}} - \sqrt{k_d}}} \cdots \cdots \cdots \underbrace{w^{(j,\sqrt{k_d})}}_{\in \mathbb{R}^{\sqrt{k_d}}} \underbrace{0 \cdots \cdots 0}_{\in \mathbb{R}^{\sqrt{\frac{p_{d-1}}{k_d}} - \sqrt{k_d}}} \underbrace{0 \cdots \cdots \cdots \cdots 0}_{\in \mathbb{R}^{p_{d-1} - \sqrt{p_{d-1}}}}.$$

Correspondingly, the stride size is $\frac{s_d^2}{k_d}$ on average and we have $\|W_d\|_2 \leq \frac{k_d}{s_d}$ if $\|w^{(j,i)}\|_2 = 1$ for all $i, j$; see Appendix F for details. This is equivalent to permuting the columns of $W_d$ generated as in (8) by vectorizing the matrix filters in order to validate the convolution of the filters with all patches of the matrix input.

**Remark 3.** A more practical scenario for CNNs is when a network has a few fully connected layers after the convolutional layers. Suppose we have $D_C$ convolutional layers and $D_F$ fully connected layers. From the analysis in Corollary 2, when $s_d = k_d$ for convolutional layers and $\|W_d\|_2 = 1$ for fully connected layers, we have that the overall ERC satisfies $\widetilde{\mathcal{O}}\left( \frac{R \cdot \sqrt{D_C k^2 + D_F pr}}{\gamma \sqrt{m}} \right)$.

## 4.2 RESNETS WITH STRUCTURED WEIGHT MATRICES

Residual networks (ResNets) (He et al., 2016) is one of the most powerful architectures that allows training of tremendously deep networks. Then we denote the class of ResNets with bounded weight matrices $\mathcal{V}_D = \{V_d \in \mathbb{R}^{p_d \times q_d}\}_{d=1}^D$, $\mathcal{U}_D = \{U_d \in \mathbb{R}^{q_d \times p_{d-1}}\}_{d=1}^D$ as

$$\mathcal{F}_{D, \|\cdot\|}^{\text{RN}} = \left\{ f\left(\mathcal{V}_D, \mathcal{U}_D, x\right) \in \mathbb{R}^{p_D} \mid \|V_d\| \leq B_{V_d}, \|U_d\| \leq B_{U_d} \right\}, \tag{9}$$

Given an input $x \in \mathbb{R}^{p_0}$, the output of a $D$-layer ResNet is defined as $f\left(\mathcal{V}_D, \mathcal{U}_D, x\right) = f_{V_D, U_D}\left(\cdots f_{V_1, U_1}(x)\right) \in \mathbb{R}^{p_D}$, where $f_{V_d, U_d}(x) = \sigma\left(V_d \cdot \sigma\left(U_d x\right) + x\right)$. For any two layers $i, j \in [D]$ and input $x$, we denote $J_{i:j}^x$ as the Jacobian from layer $i$ to layer $j$, i.e., $f_{V_i, U_j}\left(\cdots f_{V_i, U_i}(x)\right) = J_{i:j}^x \cdot x$, and $B_{i:j,2}^{\text{Jac},x}$ as an upper bound of the norm of Jacobian for input $x$ over the parameter, i.e., $\sup_{\mathcal{W}_D} \|J_{i,i}^x\|_2 \leq B_{i:j,2}^{\text{Jac},x}$. We then provide an upper bound of the ERC for ResNets in the following corollary. The proof is provided in Appendix E.

**Corollary 3.** Let $g_\gamma$ be a $\frac{1}{\gamma}$-Lipschitz and bounded loss function, i.e., $|g_\gamma| \leq b$, and $\mathcal{F}_{D, \|\cdot\|_2}^{\text{RN}}$ be the ResNets defined in (9) with $p_d = p$ and $q_d = q$ for all $d \in [D]$, $B_{\backslash d, 2}^{\text{Jac}} = \max_{d \in [D], x \in \mathcal{X}_m}$ $B_{1:(d-1),2}^{\text{Jac},x} B_{(d+1):D,2}^{\text{Jac},x}$, and $C^{\text{Net}} = \frac{B_{\backslash d,2}^{\text{Jac}} \max_d \left(B_{V_d,2} + B_{U_d,2}\right) R \sqrt{m/q}}{\sup_{f \in \mathcal{F}_{D, \|\cdot\|_2}, x \in \mathcal{X}_m} g_\gamma\left(f(\mathcal{V}_D, \mathcal{V}_D, x)\right)}$. Then the ERC satisfies

$$\mathcal{R}_m\left(\mathcal{G}_\gamma\left(\mathcal{F}^{\mathrm{RN}}_{D,\|\cdot\|_2}\right)\right) = \mathcal{O}\left(\min\left\{\frac{R\cdot\prod_{d=1}^D\left(B_{V_d,2}B_{U_d,2}+1\right)}{\gamma}, b\right\}\cdot\sqrt{\frac{Dpq\cdot\log C^{\mathrm{Net}}}{m}}\right).$$

Compared with the $D$-layer networks without shortcuts (1), ResNets have a stronger dependence on the input due to the shortcuts structure, which leads to $(B_{V_d,2}B_{U_d,2}+1)$ dependence for each layer. When $B_{V_d,2}$ and $B_{U_d,2}$ are of order $1/\sqrt{D}$, we still have $\prod_{d=1}^D\left(B_{V_d,2}B_{U_d,2}+1\right) = \mathcal{O}(1)$. This partially explains the observation in practice that ResNets have good performance when the weight matrices have relatively small scales. Also note that to achieve the same bound for $\mathcal{R}_m\left(\mathcal{G}_\gamma\left(\mathcal{F}^{\mathrm{RN}}_{D,\|\cdot\|_{\mathrm{F}}}\right)\right)$, we require $B_{V_d,\mathrm{F}}, B_{U_d,\mathrm{F}} \leq c$, which leads to a much smaller parameter space than the space corresponding to $B_{V_d,2}, B_{U_d,2} \leq c$ for the same $c$.

## 4.3 HYPERSPHERICAL NETWORKS

We also consider the hyperspherical networks (SphereNets) (Liu et al., 2017b), which demonstrate improved performance than the vanilla DNNs. In specific, the SphereNets has the same architecture with DNNs defined in (1), except that the weight matrix can be viewed as $\widetilde{W}_d = S_{W_d}W_d$, where $S_{W_d}$ is a diagonal matrix with the $i$-th diagonal entries being the Euclidean norm of the $i$-th row of $W_d$. Note that we do not normalize the input $x$ as in (Liu et al., 2017b) for ease of the discussion. A direct result of applying Theorem 1 implies that $\mathcal{R}_m\left(\mathcal{G}_\gamma\left(\mathcal{F}^{\mathrm{SN}}_{D,\|\cdot\|_2}\right)\right) = \widetilde{\mathcal{O}}(R\cdot\prod_{d=1}^D B_{\widetilde{W}_d,2}\cdot\sqrt{Dpr}/\left(\gamma\sqrt{m}\right))$. Such a self-normalization architecture has a benefit that $B_{\widetilde{W}_d,2}$ is small (close to 1) in general when the weights are spread out. In addition, it has lower computational costs than the weight normalization based on the spectral norm directly, and improved empirical results over batch normalization have been observed (Liu et al., 2017b;a).

## 4.4 EXTENSION TO WIDTH-CHANGE OPERATIONS

Change the width for certain layers is a widely used operation, e.g., for CNNs and ResNets, which can be viewed as a linear transformation in many cases. In specific, denote $x^{\{d\}} \in \mathbb{R}^{p_d}$ as the output of the $d$-th layer. Then we can use a transformation matrix $T_d \in \mathbb{R}^{p_{d+1}\times p_d}$ to denote the operation to change the dimension between the output of the $d$-th layer and the input of the $(d+1)$-th layer as $f_{W_{d+1}}\left(x^{\{d\}}\right) = \sigma\left(W_{d+1}T_d x^{\{d\}}\right)$. Denote the set of layers with width changes by $\mathcal{I}_T \subseteq [D]$. Combining with Theorem 1, we have that the ERC satisfies $\mathcal{R}_m\left(\mathcal{G}_\gamma\left(\mathcal{F}_{D,\|\cdot\|_2}\right)\right) = \widetilde{\mathcal{O}}\left(\frac{R\cdot\Pi_{d=1}^D B_{d,2}\cdot\Pi_{t\in\mathcal{I}_T}\|T_t\|_2\cdot\sqrt{Dpr}}{\gamma\sqrt{m}}\right)$. Next, we illustrate several popular examples to show that $\Pi_{t\in\mathcal{I}_T}\|T_t\|_2$ is a size independent constant. We refer to Goodfellow et al. (2016) for more operations of changing the width.

**Width Expansion.** Two popular types of width expansion are padding and $1\times1$ convolution. For ease of discussion, suppose $p_{d+1} = s\cdot p_d$ for some positive integer $s \geq 1$. Taking padding with 0 as an example, where we plug in $(s-1)$ zeros before each entry of $x^{\{d\}}$, which is equivalent to setting $T_d \in \mathbb{R}^{sp_d\times p_d}$ with $(T_d)_{ij} = 1$ if $i = js$, and $(T_d)_{ij} = 0$ otherwise. This implies that $\|T_d\|_2 = 1$.

For $1\times1$ convolution, suppose that the convolution features are $\{c_1,\ldots,c_s\}$. Then we expand width by performing convolution (essentially entry-wise product) using $s$ features respectively. This is equivalent to setting $T_d \in \mathbb{R}^{sp_d\times p_d}$ with $(T_d)_{ij} = c_k$ if $i = j+(k-1)s$ for $k \in [s]$, and $(T_d)_{ij} = 0$ otherwise. It implies that $\|T_d\|_2 = \sqrt{\sum_{i=1}^s c_i^2}$. When $\sum_{i=1}^s c_i^2 \leq 1$, we have $\|T_d\|_2 \leq 1$.

**Width Reduction.** Two popular types of width reduction are average pooling and max pooling. Suppose $p_{d+1} = \frac{p_d}{s}$ is an integer for some positive integer $s$. For average pooling, we pool each nonoverlapping $s$ features into one feature. This is equivalent to setting $T_d \in \mathbb{R}^{\frac{p_d}{s}\times p_d}$ with $(T_d)_{ij} = 1/s$ if $j = (i-1)s + k$ for $k \in [s]$, and $(T_d)_{ij} = 0$ otherwise. This implies that $\|T_d\|_2 = \sqrt{1/s}$.

For max pooling, we choose the largest entry in each nonoverlapping feature segment of length $s$. Denote the set $I_s = \{(i-1)\times s + 1,\ldots, i\cdot s\}$. This is equivalent to setting $T_d \in \mathbb{R}^{\frac{p_d}{s}\times p_d}$ with $(T_d)_{ij} = 1$ if $|(x^{\{d\}})_j| \geq |(x^{\{d\}})_k| \ \forall\, k \in I_s, k \neq j$, and $(T_d)_{ij} = 0$ otherwise. This implies that $\|T_d\|_2 = 1$. For pooling with overlapping features, similar results hold.

### 4.5 NUMERICAL EVALUATION

To better illustrate the difference between our result and existing ones, we demonstrate some comparison results in Figure 1 using real data. In specific, we train a simplified VGG19-net (Simonyan & Zisserman, 2014) using $3 \times 3$ convolution filters (with unit norm constraints) on the CIFAR-10 dataset (Krizhevsky & Hinton, 2009). We first compare with the capacity terms in Bartlett et al. (2017) (Bound1), Neyshabur et al. (2017) (Bound2), and Golowich et al. (2017) (Bound3) by ignoring the common factor $\frac{R}{\gamma\sqrt{m}}$ as follows:

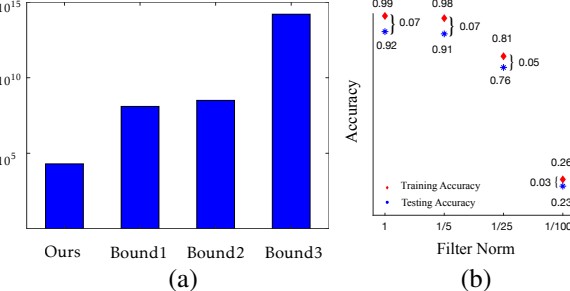

- Ours: $\Pi_{d=1}^{D} B_{d,2} \sqrt{k \sum_{d=1}^{D} n_d}$

- Bound1: $\Pi_{d=1}^{D} B_{d,2} \left( \sum_{d=1}^{D} \frac{B_{d,2\to1}^{2/3}}{B_{d,2}^{2/3}} \right)^{3/2}$

- Bound2: $\Pi_{d=1}^{D} B_{d,2} \sqrt{D^2 p \sum_{d=1}^{D} \frac{p_d B_{d,\mathrm{F}}^2}{B_{d,2}^2}}$

- Bound3: $\Pi_{d=1}^{D} B_{d,\mathrm{F}} \sqrt{D}$

Figure 1: Panel (a) shows comparison results for the same VGG19 network trained on CIFAR10 using unit norm filters. The vertical axis the corresponding bounds in the logarithmic scale. Panel (b) shows the training accuracy (red diamond), testing accuracy (blue cross), and the empirical generalization error using different scales of the filters listed on the horizontal axes.

Note that since we may have more filters $n_d$ than their dimension $k$, we do not assume orthogonality here. Thus we simply use the upper bounds of norms $B_d$ rather than the form as in Table 2. Following the analysis of Theorem 1, we have $\sqrt{k \sum_{d=1}^{D} n_d}$ dependence rather than $\sqrt{Dpr}$ as $k \sum_{d=1}^{D} n_d$ is the total number free parameter for CNNs, where $n_d$ is the number of filters at $d$-th layer. Also note that we ignore the logarithms factors in all bounds for simplicity and their empirical values are small constants compared with the the dominating terms.

For the same network and corresponding weight matrices, we see from Figure 1 (a) that our result ($10^4 \sim 10^5$) is significantly smaller than Bartlett et al. (2017); Neyshabur et al. (2017) ($10^8 \sim 10^9$) and Golowich et al. (2017) ($10^{14} \sim 10^{15}$). As we have discussed, our bound benefits from tighter dependence on the dimensions. Note that $k \sum_{d=1}^{D} n_d$ is approximately of order $Dk^2$, which is significantly smaller than $\left( \sum_{d=1}^{D} B_{d,2\to1}^{2/3} / B_{d,2}^{2/3} \right)^3$ in Bartlett et al. (2017) and $D^2 p \sum_{d=1}^{D} p_d B_{d,\mathrm{F}}^2 / B_{d,2}^2$ in Neyshabur et al. (2017) (both are of order $D^3 pr$). In addition, this verifies that spectral dependence is significantly tighter than Frobenius norm dependence in Golowich et al. (2017). Further, we show the training accuracy, testing accuracy, and the empirical generalization error using different scales on the norm of the filters in Figure 1 (b). We see that the generalization errors decrease when the norm of filters decreases. However, note that when the norms are too small, the accuracies drop significantly due to a potentially much smaller parameter space. Thus, the scales (norms) of the weight matrices should be nether too large (induce large generalization error) nor too small (induce low accuracy) and choosing proper scales is important in practice as existing works have shown. On the other hand, this also support our claim that when $\mathcal{R}_m \left( \mathcal{G}_\gamma \left( \mathcal{F}_{D, \|\cdot\|_{\mathrm{F}}} \right) \right)$ (or other existing bound) attains the same order with our $\mathcal{R}_m \left( \mathcal{G}_\gamma \left( \mathcal{F}_{D, \|\cdot\|_2} \right) \right)$, we have better training/testing performance. Further experimental result that compare our bound (Corollary 1) with (Zhou & Feng, 2018; Arora et al., 2018) in the bounded output case is provided in Appendix A.1.

We want to remark that all numerical evaluations are empirical estimation of the generalization bounds, rather than their exact values. This is because all existing bounds requires to take uniform bounds of some quantities on parameters or the supremum value over the entire space, which is empirically not accessible. For example, in the case that when it involve the upper/lower bound of quantities (norm, rank, or other parameters) depending on weight matrices, theoretically we should take the values of their upper/lower bounds (this leads to worse empirical bounds) rather than estimating them from the training process; or in the case that the bounds involve some quantities depending on the supremum over the entire parameter space, numerical evaluations cannot exhaust the entire parameter space to reach the supremum (Bartlett et al., 2017; Golowich et al., 2017; Neyshabur et al., 2015; 2017; Zhou & Feng, 2018; Arora et al., 2018). Our experiments here (including Appendix A) cannot avoid such restrictions, but the comparison is fair across various bounds as they are obtained from the same training process.

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

# A EXTENDED NUMERICAL RESULTS

## A.1 COMPARISON WITH EXISTING RESULTS WITH BOUNDED OUTPUT

We compared our norms based bound with several other norm based results in Section 4.5. In this section, we further compare our result with bounded output in Corollary 1 with Zhou & Feng (2018); Arora et al. (2018). Analogous to Section 4.5, we train a simplified VGG19-net (Simonyan & Zisserman, 2014) using $3 \times 3$ convolution filters (with unit norm constraints) on the CIFAR-10 dataset (Krizhevsky & Hinton, 2009). We compare with the capacity terms in Zhou & Feng (2018) (Bound 1) and Arora et al. (2018) (Bound 2) by ignoring the factor $\frac{R}{\gamma\sqrt{m}}$ as follows:

- Ours: $b\sqrt{k\sum_{d=1}^{D} n_d}$. Note that $b = 1$ in this case.

- Bound1: $\sqrt{Dk\sum_{d=1}^{D} \mathrm{rank}(W_d) \cdot n_d}$. Note that their activation functions for the intermediate layers are sigmoid and the last layer is softmax, with squared error, which essentially has a bounded output. We also take the last layer as convolution layer for ease of discussion.

- Bound2: $\max_{x \in \mathcal{X}_m} \|f(\mathcal{W}_D, x)\|_2 \cdot CD\beta\sqrt{\sum_{d=1}^{D} \frac{\lceil k/s \rceil^2}{\mu_d^2 \mu_{d\to}^2}}$, where $C = \Omega(1)$ is the activation contraction, $\beta = \Omega(1)$ is the well-distributed Jacobian, $\mu_d = \mathcal{O}(1/\sqrt{p})$ is the layer cushion, and $\mu_{d\to} = \mathcal{O}(1/\sqrt{p})$ is the interlayer cushion. See more details in Arora et al. (2018).

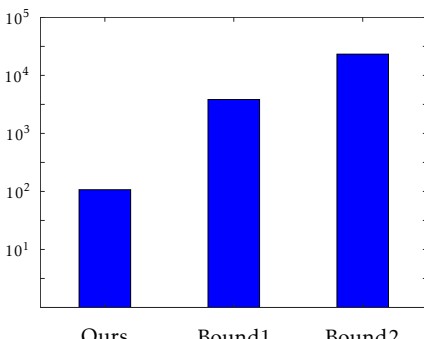

Figure 2: Comparison results for the same VGG19 network trained on CIFAR10. The vertical axis is the corresponding bounds in the logarithmic scale.

The resulting bounds on the trained networks are provided in Figure 2. We observe that our bound is smaller than Zhou & Feng (2018); Arora et al. (2018) by at least an order of magnitude. Specifically, our bound is of order $\approx 10^2$, while Zhou & Feng (2018) result in a bound of order $> 10^3$ and Arora et al. (2018) result in a bound of order $> 10^4$. This coincide with our discussion in Section 3.2 and allows us to obtain non-vacuous bound (generalization bound $< 1$) with moderate training sample sizes (e.g., $m = \Omega(10^4)$). Also note that compared with the norm based based results in Figure 1, the results here based on the bounded output are significantly tighter in general. We regard this as a trade-off between the looser norm based bounds that allow more explicit interpretability in terms of the weight matrix structures and the tighter bounds that are more obscure in terms of dependence on the network structures. A result that have both merits are desired as a future direction.

## A.2 COMPARISON BETWEEN $B_{\backslash d,2}^{\mathrm{Jac}}$ AND $\prod_d B_{d,2}$

We demonstrate the empirical difference between $B_{\backslash d,2}^{\mathrm{Jac}}$ and $\prod_d B_{d,2}$. Using the same setting of the network and dataset as above, we provide the empirical distribution of $B_{\backslash d,2}^{\mathrm{Jac}}$ and $\prod_d B_{d,2}$ over the training set using different random initializations of weight matrices, which is provided in Figure 3. We can observe that the values of $B_{\backslash d,2}^{\mathrm{Jac}}$ are approximately 2 orders smaller than the values of $\prod_d B_{d,2}$, which support our claim that $B_{\backslash d,2}^{\mathrm{Jac}}$ is a tighter quantification then $\prod_d B_{d,2}$.

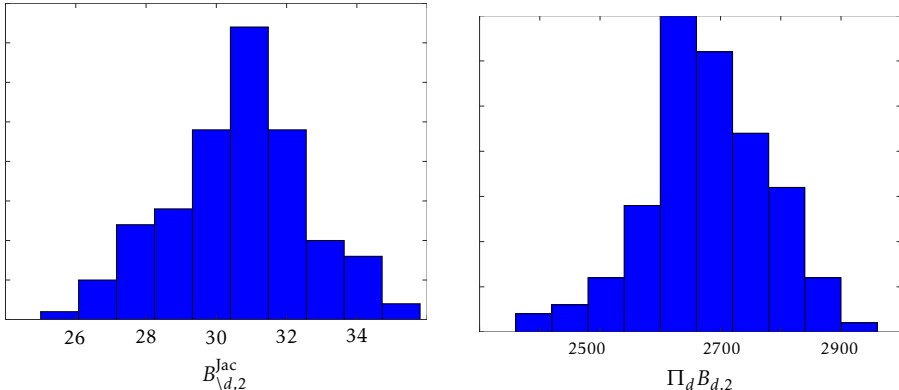

Figure 3: Empirical distribution of $B^{\text{Jac}}_{\backslash d,2}$ and $\prod_d B_{d,2}$ for the same VGG19 network trained on CIFAR10.

### A.3 Dependence of $B^{\text{Jac}}_{\backslash d,2}$ and $\prod_d B_{d,2}$ on Depth

We further provide an empirical evaluation to see how strong the quantities $B^{\text{Jac}}_{\backslash d,2}$ and $\prod_d B_{d,2}$ depend on the depth. Note that we use $d$ as the variable for depth. Using the same setting as above, we provide the magnitude of $\log B^{\text{Jac}}_{\backslash d,2}$ and $\log \prod_d B_{d,2}$ in Figure 4. We also provide the plots for $\log d$ and $\log d^2$ as reference. We can observe that $\log \prod_d B_{d,2}$ has an approximately linear dependence on the depth, which matches with our intuition. In terms of $\log B^{\text{Jac}}_{\backslash d,2}$, we can see that it has a significantly slower increasing rate than $\log \prod_d B_{d,2}$. Compared with the reference plots $\log d$ and $\log d^2$, we can observe even a slower rate than $\log d^2$. This further indicates that $\log B^{\text{Jac}}_{\backslash d,2}$ may has a dependence slower than some low degree of $\text{poly}(d)$.

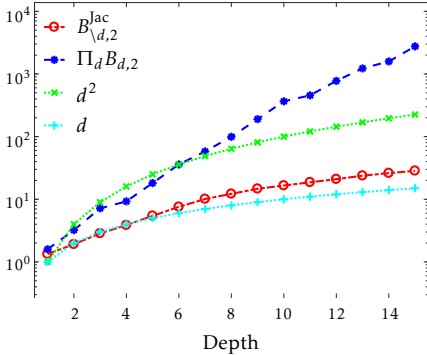

Figure 4: Comparison results for the dependence of $B^{\text{Jac}}_{\backslash d,2}$ and $\prod_d B_{d,2}$ on depth. The horizontal axes is the depth and the vertical axes is the values of corresponding quantities in the logarithmic scale.

## B Proof of Theorem 1

We start with some definitions of notations. Given a vector $x \in \mathbb{R}^p$, we denote $x_i$ as the $i$-th entry, and $x_{i:j}$ as a sub-vector indexed from $i$-th to $j$-th entries of $x$. Given a matrix $A \in \mathbb{R}^{n \times m}$, we denote $A_{ij}$ as the entry corresponding to $i$-th row and $j$-th column, $A_{i*}$ ($A_{*i}$) as the $i$-th row (column), $A_{\mathcal{I}_1 \mathcal{I}_2}$ as a submatrix of $A$ indexed by the set of rows $\mathcal{I}_1 \subseteq [n]$ and columns $\mathcal{I}_2 \subseteq [m]$. Given two real values $a, b \in \mathbb{R}^+$, we write $a \lesssim (\gtrsim) b$ if $a \leq (\geq) cb$ for some generic constant $c > 0$.

Our analysis is based on the characterization of the Lipschitz property of a given function on both input and parameters. Such an idea can potentially provide tighter bound on the model capacity in terms of these Lipschitz constants and the number of free parameters, including other architectures

of DNNs. We first provide an upper bound for the Lipschitz constant of $f(\mathcal{W}_D, x)$ in terms of the input $x$.

**Lemma 1.** Given $\mathcal{W}_D$, for any $f(\mathcal{W}_D, \cdot) \in \mathcal{F}_{D,\|\cdot\|_2}$ and $x_1, x_2 \in \mathbb{R}^{p_0}$, we have

$$\|f(\mathcal{W}_D, x_1) - f(\mathcal{W}_D, x_2)\|_2 \leq \|x_1 - x_2\|_2 \cdot \prod_{d=1}^{D} B_d.$$

*Proof.* We prove by induction. Specifically, we have

$$
\begin{aligned}
\|f(\mathcal{W}_D, x_1) - f(\mathcal{W}_D, x_2)\|_2 &= \|\sigma(W_D f(\mathcal{W}_{D-1}, x_1)) - \sigma(W_D f(\mathcal{W}_{D-1}, x_2))\|_2 \\
&\overset{(i)}{\leq} \|W_D f(\mathcal{W}_{D-1}, x_1) - W_D f(\mathcal{W}_{D-1}, x_2)\|_2 \\
&\leq \|W_D\|_2 \cdot \|f(\mathcal{W}_{D-1}, x_1) - f(\mathcal{W}_{D-1}, x_2)\|_2 \\
&\leq B_D \cdot \|f(\mathcal{W}_{D-1}, x_1) - f(\mathcal{W}_{D-1}, x_2)\|_2,
\end{aligned}
$$

where $(i)$ comes from the entry-wise 1–Lipschitz continuity of $\sigma(\cdot)$. For the first layer, we have

$$
\begin{aligned}
\|f(\mathcal{W}_1, x_1) - f(\mathcal{W}_1, x_2)\|_2 &= \|\sigma(W_1 x_1) - \sigma(W_D x_2)\|_2 \leq \|W_1 x_1 - W_1 x_2\|_2 \\
&\leq B_1 \cdot \|x_1 - x_2\|_2.
\end{aligned}
$$

By repeating the argument above, we complete the proof. $\square$

Next, we provide an upper bound for the Lipschitz constant of $f(\mathcal{W}_D, x)$ in terms of the parameters $\mathcal{W}_D$.

**Lemma 2.** Given any $x \in \mathbb{R}^{p_0}$ satisfying $\|x\|_2 \leq R$, for any $f(\mathcal{W}_D, x), f\left(\widetilde{\mathcal{W}}_D, x\right) \in \mathcal{F}_{D,\|\cdot\|_2}$ with $\mathcal{W}_D = \{W_d\}_{d=1}^{D}$ and $\widetilde{\mathcal{W}}_D = \left\{\widetilde{W}_d\right\}_{d=1}^{D}$, and denote $B_{\backslash d,2}^{\mathrm{Jac},x} = \max_{d \in [D]} B_{1:(d-1),2}^{\mathrm{Jac},x} B_{(d+1):D,2}^{\mathrm{Jac},x}$, then we have

$$\left\|f(\mathcal{W}_D, x) - f\left(\widetilde{\mathcal{W}}_D, x\right)\right\|_2 \leq B_{\backslash d,2}^{\mathrm{Jac},x} \cdot R\sqrt{D} \sqrt{\sum_{d=1}^{D} \|W_d - \widetilde{W}_d\|_{\mathrm{F}}^2}.$$

*Proof.* Given $x$ and two sets of weight matrices $\{W_d\}_{d=1}^{D}, \left\{\widetilde{W}_d\right\}_{d=1}^{D}$, we have

$$
\begin{aligned}
&\left\|f_{W_D}\left(f_{W_{D-1}}\left(\cdots f_{W_1}(x)\right)\right) - f_{\widetilde{W}_D}\left(f_{\widetilde{W}_{D-1}}\left(\cdots f_{\widetilde{W}_1}(x)\right)\right)\right\|_2 \\
&\overset{(i)}{=} \left\|\sum_{d=1}^{D} f_{W_D}\left(\cdots f_{W_{d+1}}\left(f_{\widetilde{W}_d}\left(\cdots f_{\widetilde{W}_1}(x)\right)\right)\right) - f_{W_D}\left(\cdots f_{W_d}\left(f_{\widetilde{W}_{d-1}}\left(\cdots f_{\widetilde{W}_1}(x)\right)\right)\right)\right\|_2 \\
&\leq \sum_{d=1}^{D} \left\|f_{W_D}\left(\cdots f_{W_{d+1}}\left(f_{\widetilde{W}_d}\left(\cdots f_{\widetilde{W}_1}(x)\right)\right)\right) - f_{W_D}\left(\cdots f_{W_d}\left(f_{\widetilde{W}_{d-1}}\left(\cdots f_{\widetilde{W}_1}(x)\right)\right)\right)\right\|_2 \\
&\overset{(ii)}{=} \sum_{d=1}^{D} \left\|J_{(d+1):D}^{x} \cdot f_{\widetilde{W}_d}\left(\cdots f_{\widetilde{W}_1}(x)\right) - J_{(d+1):D}^{x} \cdot f_{W_d}\left(f_{\widetilde{W}_{d-1}}\left(\cdots f_{\widetilde{W}_1}(x)\right)\right)\right\|_2 \\
&\overset{(iii)}{\leq} \sum_{d=1}^{D} B_{(d+1):D}^{\mathrm{Jac},x} \cdot \left\|\widetilde{W}_d f_{\widetilde{W}_{d-1}}\left(\cdots f_{\widetilde{W}_1}(x)\right) - W_d f_{\widetilde{W}_{d-1}}\left(\cdots f_{\widetilde{W}_1}(x)\right)\right\|_2 \\
&\leq \sum_{d=1}^{D} B_{(d+1):D}^{\mathrm{Jac},x} \cdot \left\|W_d - \widetilde{W}_d\right\|_2 \cdot \left\|f_{\widetilde{W}_{d-1}}\left(\cdots f_{\widetilde{W}_1}(x)\right)\right\|_2,
\end{aligned}
\tag{10}
$$

where $(i)$ is derived from adding and subtracting intermediate neural network functions recurrently, where $f_{W_D}\left(\cdots f_{W_{d+1}}\left(f_{\widetilde{W}_d}\left(\cdots f_{\widetilde{W}_1}(x)\right)\right)\right)$ share the same output of activation functions from $d+1$-th layer to $D$-the layer with $f_{W_D}\left(f_{W_{D-1}}\left(\cdots f_{W_1}(x)\right)\right)$, $(ii)$ is from fixing the activation

function output, and $(iii)$ is from the entry-wise 1–Lipschitz continuity of $\sigma(\cdot)$. On the other hand, for any $d \in [D]$, we further have

$$\left\| f_{W_d} \left( \cdots f_{W_1}(x) \right) \right\|_2 = \left\| J_{1:d}^x \cdot x \right\|_2 \leq B_{1:d}^{\text{Jac},x} \cdot \|x\|_2. \tag{11}$$

where $(i)$ is from the entry-wise 1–Lipschitz continuity of $\sigma(\cdot)$ and $(ii)$ is from recursively applying the same argument.

In addition, we denote $W_d = U_d V_d^\top$ and $\widetilde{W}_d = \widetilde{U}_d \widetilde{V}_d^\top$, where $U_d, V_d, \widetilde{U}_d, \widetilde{V}_d \in \mathbb{R}^{p \times r}$ and $\|U\|_2 = \|V\|_2 = \left\| \widetilde{U} \right\|_2 = \left\| \widetilde{V} \right\|_2 = \|W_d\|_2^{1/2}$. Then we have

$$\begin{aligned}
\left\| W_d - \widetilde{W}_d \right\|_2 &= \left\| U_d V_d^\top - \widetilde{U}_d \widetilde{V}_d^\top \right\|_2 \\
&= \left\| U_d V_d^\top - U_d \widetilde{V}_d^\top + U_d \widetilde{V}_d^\top - \widetilde{U}_d \widetilde{V}_d^\top \right\|_2 \\
&\leq \|U\|_2 \left\| V - \widetilde{V} \right\|_2 + \left\| \widetilde{V} \right\|_2 \left\| U - \widetilde{U} \right\|_2 \\
&\leq \|W_d\|_2^{1/2} \left( \left\| V - \widetilde{V} \right\|_{\text{F}} + \left\| U - \widetilde{U} \right\|_{\text{F}} \right).
\end{aligned} \tag{12}$$

Applying (10) recursively and combining (11) and (12), we obtain the desired result as

$$\begin{aligned}
&\left\| f_{W_D} \left( f_{W_{D-1}} \left( \cdots f_{W_1}(x) \right) \right) - f_{\widetilde{W}_D} \left( f_{\widetilde{W}_{D-1}} \left( \cdots f_{\widetilde{W}_1}(x) \right) \right) \right\|_2 \\
&\leq \sum_{d=1}^{D} B_{(d+1):D}^{\text{Jac},x} \cdot B_{1:(d-1)}^{\text{Jac},x} \cdot \|x\|_2 \cdot \|W_d\|_2^{1/2} \left( \left\| V - \widetilde{V} \right\|_{\text{F}} + \left\| U - \widetilde{U} \right\|_{\text{F}} \right) \\
&\leq B_{\backslash d,2}^{\text{Jac},x} \cdot R\sqrt{D} \cdot \max_d B_{d,2}^{1/2} \sum_{d=1}^{D} \left( \left\| V - \widetilde{V} \right\|_{\text{F}} + \left\| U - \widetilde{U} \right\|_{\text{F}} \right) \\
&\leq B_{\backslash d,2}^{\text{Jac},x} \cdot R\sqrt{2D} \cdot \max_d B_{d,2}^{1/2} \sqrt{\sum_{d=1}^{D} \left\| V - \widetilde{V} \right\|_{\text{F}}^2 + \left\| U - \widetilde{U} \right\|_{\text{F}}^2}.
\end{aligned}$$

$\square$

**Lemma 3.** Suppose $g(w, x)$ is $L_w$-Lipschitz over $w \in \mathbb{R}^h$ with $\|w\|_2 \leq K$ and $\alpha = \sup_{g \in \mathcal{G}, x \in \mathcal{X}_m} |g(w, x)|$. Then the ERC of $\mathcal{G} = \{g(w, x)\}$ satisfies

$$\mathcal{R}_m(\mathcal{G}) = \mathcal{O}\left( \frac{\alpha \sqrt{h \log \frac{K L_w \sqrt{m}}{\alpha \sqrt{h}}}}{\sqrt{m}} \right).$$

*Proof.* For any $w, \widetilde{w} \in \mathbb{R}^h$ and $\mathcal{X}_m = \{x_i\}_{i=1}^m$, we consider the matric $\Delta(g_1, g_2) = \max_{x_i \in \mathcal{X}_m} |g_1(x_i) - g_2(x_i)|$, which satisfies

$$\Delta(g_1, g_2) = \max_{x \in \mathcal{X}_m} |g_1(x) - g_2(x)| = |g(w, x) - g(\widetilde{w}, x)| \leq L_w \|w - \widetilde{w}\|_2. \tag{13}$$

Since $g$ is a parametric function with $h$ parameters, then we have the covering number of $\mathcal{G}$ under the metric $\Delta$ in (13) satisfies

$$\mathcal{N}(\mathcal{G}, \Delta, \delta) \leq \left( \frac{3 K L_w}{\delta} \right)^h.$$

Then using the standard Dudley's entropy integral bound on the ERC (Mohri et al., 2012), we have the ERC satisfies

$$\mathcal{R}_m(\mathcal{G}) \lesssim \inf_{\beta > 0} \beta + \frac{1}{\sqrt{m}} \int_{\beta}^{\sup_{g \in \mathcal{G}} \Delta(g, 0)} \sqrt{\log \mathcal{N}(\mathcal{G}, \Delta, \delta)} \, d\delta. \tag{14}$$

Since we have

$$\alpha = \sup_{g \in \mathcal{G}, x \in \mathcal{X}_m} \Delta(g, 0) = \sup_{g \in \mathcal{G}, x \in \mathcal{X}_m} |g(w, x)|.$$

Then we have

$$\mathcal{R}_m\left(\mathcal{G}\right) \lesssim \inf_{\beta>0} \beta + \frac{1}{\sqrt{m}} \int_\beta^\alpha \sqrt{h \log \frac{KL_w}{\delta}} \, d\delta$$

$$\leq \inf_{\beta>0} \beta + \alpha \sqrt{\frac{h \log \frac{KL_w}{\beta}}{m}} \stackrel{(i)}{\lesssim} \frac{\alpha \sqrt{h \log \frac{KL_w \sqrt{m}}{\alpha \sqrt{h}}}}{\sqrt{m}},$$

where $(i)$ is obtained by taking $\beta = \alpha \sqrt{h/m}$. $\qquad\qquad\qquad\square$

By definition, we have $\alpha = \sup_{f \in \mathcal{F}_{D,\|\cdot\|_2}, x \in \mathcal{X}_m} g_\gamma\left(f\left(\mathcal{W}_D, x\right)\right)$. From Lemma 1 and $\frac{1}{\gamma}$-Lipschitz continuity of $g$, we also have

$$\alpha \leq \frac{L_x R}{\gamma} \leq \frac{R \cdot \prod_{d=1}^D B_{d,2}}{\gamma}. \tag{15}$$

From Lemma 2, we have

$$L_w \leq \max_{x \in \mathcal{X}_m} B_{\backslash d,2}^{\mathrm{Jac},x} \cdot R\sqrt{2D} \cdot \max_d B_{d,2}^{1/2}.$$

Moreover, when $p_d = p$ for all $d \in [D]$, we have

$$K = \sqrt{\sum_{d=1}^D \|W_d\|_{\mathrm{F}}^2} \leq \sqrt{pD} \cdot \max_d B_{d,2}.$$

Combining the results above with Lemma 3 and $h = 2Dpr$, we have

$$\mathcal{R}_m\left(\mathcal{G}\right) \lesssim \frac{\alpha \sqrt{h \log \frac{KL_w \sqrt{m}}{\alpha \sqrt{h}}}}{\sqrt{m}}$$

$$\lesssim \frac{R \cdot \prod_{d=1}^D B_{d,2} \sqrt{Dpr \log \frac{B_{\backslash d,2}^{\mathrm{Jac}} \cdot R\sqrt{Dm/r} \cdot \max_d B_{d,2}}{\sup_{f \in \mathcal{F}_{D,\|\cdot\|_2}, x \in \mathcal{X}_m} g_\gamma(f(\mathcal{W}_D,x))}}}{\gamma \sqrt{m}}.$$

## C    PROOF OF COROLLARY 1

The analysis follows Theorem 1, except that the bound for $\alpha$ in (15) satisfies

$$\alpha \leq \min\left\{b, \frac{R \cdot \prod_{d=1}^D B_{d,2}}{\gamma}\right\},$$

since $g$ satisfies $|g| \leq b$ and $\frac{1}{\gamma}$-Lipschitz continuous. Then we have the desired result.

## D    PROOF OF COROLLARY 2

We first show that using unit norm filters for all $d \in [D]$ and $n_d \leq k_d$, we have

$$\|W_d\|_2 = \sqrt{\frac{k_d}{s_d}}, \tag{16}$$

First note that when $n_d = k_d$, due to the orthogonality of $\left\{w^{(d,j)}\right\}_{j=1}^{k_d}$, for all $i, q \in [k_d]$, $i \neq q$, we have

$$\sum_{j=1}^{k_d} \left(w_i^{(d,j)}\right)^2 = 1 \text{ and } \sum_{j=1}^{k_d} w_q^{(d,j)} \cdot w_i^{(d,j)} = 0. \tag{17}$$

When $n_d = k_d$, we have for all $i \in [p_{d-1}]$, the diagonal entries of $W_d^\top W_d$ satisfy

$$\left(W_d^\top W_d\right)_{ii} = \sum_{j=1}^{k_d} \left\|\left(W_d^{(j)}\right)_{*i}\right\|_2 = \sum_{j=1}^{k_d} \sum_{h=1}^{\frac{k_d}{s_d}} \left(w_{(i\%s_d)+(h-1)s_d}^{(d,j)}\right)^2 \stackrel{(i)}{=} \frac{k_d}{s_d}. \tag{18}$$

where $(i)$ is from (17). For the off-diagonal entries of $W_d^\top W_d$, i.e., for $i \neq q$, $i, q \in [p_d]$, we have

$$\left(W_d^\top W_d\right)_{iq} = \sum_{j=1}^{k_d} \left(W_d^{(j)}\right)_{*q}^\top \left(W_d^{(j)}\right)_{*i}$$

$$= \sum_{j=1}^{k_d} \sum_{h=1}^{\frac{k_d}{s_d}} w_{(i\%s_d)+(h-1)s_d}^{(d,j)} \cdot w_{(q\%s_d)+(h-1)s_d}^{(d,j)}$$

$$\stackrel{(i)}{=} 0, \tag{19}$$

where $(i)$ is from (17). Combining (18) and (19), we have that $W_d^\top W_d$ is a diagonal matrix with

$$\left\|W_d^\top W_d\right\|_2 = \frac{k_d}{s_d} \implies \|W_d\|_2 = \sqrt{\frac{k_d}{s_d}}.$$

For $n_d < n_k$, we have that $W_d$ is a row-wise submatrix of that when $n_d = k_d$, denoted as $\widetilde{W}_d$. Let $S \in \mathbb{R}^{\frac{n_d k_d}{s_d} \times p_d}$ be a row-wise submatrix of an identity matrix corresponding to sampling the row of $W_d$ to form $\widetilde{W}_d$. Then we have that (16) holds, and since

$$\left\|\widetilde{W}_d\right\|_2 = \sqrt{\left\|S \cdot W_d W_d^\top \cdot S^\top\right\|_2^2} = \sqrt{\frac{k_d}{s_d}}.$$

Suppose $k_1 = \cdots = k_D = k$ for ease of discussion. Then following the same argument as in the proof of Theorem 1 and Lemma 3, we have

$$\alpha = \sup_{f \in \mathcal{F}_{D,\|\cdot\|_2}, x \in \mathcal{X}_m} g_\gamma\left(f\left(\mathcal{W}_D, x\right)\right) \leq \frac{R \cdot \prod_{d=1}^D B_{d,2}}{\gamma} = \frac{R \cdot \prod_{d=1}^D \sqrt{\frac{k}{s_d}}}{\gamma},$$

$$L_w \leq \max_{x \in \mathcal{X}_m} B_{\backslash d,2}^{\text{Jac},x} \cdot R\sqrt{2Dk/s},$$

$$K = \sqrt{\sum_{d=1}^D \sum_{j=1}^{n_d} \left\|w^{(d,j)}\right\|_2^2} = \sqrt{\sum_{d=1}^D n_d}, \text{ and}$$

$$h = k \sum_{d=1}^D n_d.$$

Using the fact that the number of parameters in each layer is no more than $kn_d$ rather than $2pr$, we have

$$\mathcal{R}_m\left(\mathcal{G}\right) \lesssim \frac{\alpha\sqrt{h \log \frac{KL_w\sqrt{m}}{\alpha\sqrt{h}}}}{\sqrt{m}}$$

$$\lesssim \frac{R \cdot \prod_{d=1}^D \sqrt{\frac{k}{s_d}} \cdot \sqrt{k \sum_{d=1}^D n_d \log \frac{B_{\backslash d,2}^{\text{Jac}} \cdot R\sqrt{Dm/s}}{\sup_{f \in \mathcal{F}_{D,\|\cdot\|_2}, x \in \mathcal{X}_m} g_\gamma(f(\mathcal{W}_D, x))}}}{\gamma\sqrt{m}}.$$

## E   PROOF OF COROLLARY 3

The analysis is analogous to the proof for Theorem 1, but with different construction of the intermediate results. We first provide an upper bound for the Lipschitz constant of $f\left(\mathcal{V}_D, \mathcal{U}_D, x\right)$ in terms of $x$.

**Lemma 4.** Given $\mathcal{V}_D$ and $\mathcal{U}_D$, for any $f\left(\mathcal{V}_D, \mathcal{U}_D, \cdot\right) \in \mathcal{F}_{D, \|\cdot\|_2}$ and $x_1, x_2 \in \mathbb{R}^{p_0}$, we have

$$\left\|f\left(\mathcal{V}_D, \mathcal{U}_D, x_1\right) - f\left(\mathcal{V}_D, \mathcal{U}_D, x_2\right)\right\|_2 \leq \left\|x_1 - x_2\right\|_2 \cdot \Pi_{d=1}^{D}\left(B_{U_d, 2} B_{V_d, 2} + 1\right). \tag{20}$$

*Proof.* Consider the ResNet layer, for any $x_1, x_2 \in \mathbb{R}^k$, we have

$$
\begin{aligned}
&\left\|f\left(\mathcal{V}_D, \mathcal{U}_D, x_1\right) - f\left(\mathcal{V}_D, \mathcal{U}_D, x_2\right)\right\|_2 \\
&= \left\|f_{V_D, U_D}\left(\cdots f_{V_1, U_1}\left(x_1\right)\right) - f_{V_D, U_D}\left(\cdots f_{V_1, U_1}\left(x_2\right)\right)\right\|_2 \\
&= \left\|\sigma\left(V_D \cdot \sigma\left(U_D \cdot f_{V_{D-1}, U_{D-1}}\left(\cdots f_{V_1, U_1}\left(x_1\right)\right)\right) + f_{V_{D-1}, U_{D-1}}\left(\cdots f_{V_1, U_1}\left(x_1\right)\right)\right) \right. \\
&\qquad \left. - \sigma\left(V_D \cdot \sigma\left(U_D \cdot f_{V_{D-1}, U_{D-1}}\left(\cdots f_{V_1, U_1}\left(x_2\right)\right)\right) + f_{V_{D-1}, U_{D-1}}\left(\cdots f_{V_1, U_1}\left(x_2\right)\right)\right)\right\|_2 \\
&\overset{(i)}{\leq} \left\|V_D \cdot \sigma\left(U_D \cdot f_{V_{D-1}, U_{D-1}}\left(\cdots f_{V_1, U_1}\left(x_1\right)\right)\right) - V_D \cdot \sigma\left(U_D \cdot f_{V_{D-1}, U_{D-1}}\left(\cdots f_{V_1, U_1}\left(x_2\right)\right)\right)\right\|_2 \\
&\qquad + \left\|f_{V_{D-1}, U_{D-1}}\left(\cdots f_{V_1, U_1}\left(x_1\right)\right) - f_{V_{D-1}, U_{D-1}}\left(\cdots f_{V_1, U_1}\left(x_2\right)\right)\right\|_2 \\
&\overset{(ii)}{\leq} \left(\left\|V_D\right\|_2 \left\|U_D\right\|_2 + 1\right) \cdot \left\|f_{V_{D-1}, U_{D-1}}\left(\cdots f_{V_1, U_1}\left(x_1\right)\right) - f_{V_{D-1}, U_{D-1}}\left(\cdots f_{V_1, U_1}\left(x_2\right)\right)\right\|_2,
\end{aligned}
$$

where $(i)$ is the fact that $\sigma$ is 1–Lipschitz, and $(ii)$ is from repeating the arguments of $(i)$ and $(ii)$. By recursively applying the argument above, we have the desired result. □

Next, we provide an upper bound for the Lipschitz constant of $f\left(\mathcal{V}_D, \mathcal{U}_D, x\right)$ in terms of $\mathcal{V}_D$ and $\mathcal{U}_D$.

**Lemma 5.** Given any $x \in \mathbb{R}^{p_0}$ satisfying $\|x\|_2 \leq R$, for any $f\left(\mathcal{V}_D, \mathcal{U}_D, x\right), f\left(\widetilde{\mathcal{V}}_D, \widetilde{\mathcal{U}}_D, x\right) \in \mathcal{F}_{D, \|\cdot\|_2}$ with $\mathcal{V}_D = \{V_d\}_{d=1}^{D}, \mathcal{U}_D = \{U_d\}_{d=1}^{D}, \widetilde{\mathcal{V}}_D = \left\{\widetilde{V}_d\right\}_{d=1}^{D}$, and $\widetilde{\mathcal{U}}_D = \left\{\widetilde{U}_d\right\}_{d=1}^{D}$, and denote $B_{\backslash d, 2}^{\mathrm{Jac}, x} = \max_{d \in [D]} B_{1:(d-1), 2}^{\mathrm{Jac}, x} B_{(d+1):D, 2}^{\mathrm{Jac}, x}$, then we have

$$
\begin{aligned}
&\left\|f\left(\mathcal{V}_D, \mathcal{U}_D, x\right) - f\left(\widetilde{V}_D, \widetilde{U}_D, x\right)\right\|_2 \\
&\leq B_{\backslash d, 2}^{\mathrm{Jac}, x} \max_{d}\left(B_{V_d, 2} + B_{U_d, 2}\right) R\sqrt{2D} \cdot \sqrt{\sum_{d=1}^{D}\left\|V_D - \widetilde{V}_D\right\|_{\mathrm{F}}^2 + \sum_{d=1}^{D}\left\|U_D - \widetilde{U}_D\right\|_{\mathrm{F}}^2}.
\end{aligned}
$$

*Proof.* Given $x$ and two sets of weight matrices $\{W_d\}_{d=1}^{D}$, $\left\{\widetilde{W}_d\right\}_{d=1}^{D}$, we have

$$
\left\| f_{V_D,U_D}\left(f_{V_{D-1},U_{D-1}}\left(\cdots f_{V_1,U_1}(x)\right)\right) - f_{\widetilde{V}_D,\widetilde{U}_D}\left(f_{\widetilde{V}_{D-1},\widetilde{U}_{D-1}}\left(\cdots f_{\widetilde{V}_1,\widetilde{U}_1}(x)\right)\right) \right\|_2
$$

$$
\leq \left\| \sum_{d=1}^{D} f_{V_D,U_D}\left(\cdots f_{V_{d+1},U_{d+1}}\left(f_{\widetilde{V}_d,\widetilde{U}_d}(\cdots)\right)\right) - f_{V_D,U_D}\left(\cdots f_{V_{d+1},U_{d+1}}\left(f_{\widetilde{V}_d,U_d}(\cdots)\right)\right) \right.
$$

$$
\left. + f_{V_D,U_D}\left(\cdots f_{V_{d+1},U_{d+1}}\left(f_{\widetilde{V}_d,\widetilde{U}_d}(\cdots)\right)\right) - f_{V_D,U_D}\left(\cdots f_{V_d,U_d}\left(f_{\widetilde{V}_{d-1},\widetilde{U}_{d-1}}(\cdots)\right)\right) \right\|_2
$$

$$
\leq \sum_{d=1}^{D} \left\| f_{V_D,U_D}\left(\cdots f_{V_{d+1},U_{d+1}}\left(f_{\widetilde{V}_d,\widetilde{U}_d}(\cdots)\right)\right) - f_{V_D,U_D}\left(\cdots f_{V_{d+1},U_{d+1}}\left(f_{\widetilde{V}_d,U_d}(\cdots)\right)\right) \right\|_2
$$

$$
+ \sum_{d=1}^{D} \left\| f_{V_D,U_D}\left(\cdots f_{V_{d+1},U_{d+1}}\left(f_{\widetilde{V}_d,U_d}(\cdots)\right)\right) - f_{V_D,U_D}\left(\cdots f_{V_d,U_d}\left(f_{\widetilde{V}_{d-1},\widetilde{U}_{d-1}}(\cdots)\right)\right) \right\|_2
$$

$$
= \sum_{d=1}^{D} \left\| J_{(d+1):D}^{x} \cdot f_{\widetilde{V}_d,\widetilde{U}_d}\left(f_{\widetilde{V}_{d-1},\widetilde{U}_{d-1}}(\cdots)\right) - J_{(d+1):D}^{x} \cdot f_{\widetilde{V}_d,U_d}\left(f_{\widetilde{V}_{d-1},\widetilde{U}_{d-1}}(\cdots)\right) \right\|_2
$$

$$
+ \sum_{d=1}^{D} \left\| J_{(d+1):D}^{x} \cdot f_{\widetilde{V}_d,U_d}\left(f_{\widetilde{V}_{d-1},\widetilde{U}_{d-1}}(\cdots)\right) - J_{(d+1):D}^{x} \cdot f_{V_d,U_d}\left(f_{\widetilde{V}_{d-1},\widetilde{U}_{d-1}}(\cdots)\right) \right\|_2
$$

$$
\leq \sum_{d=1}^{D} B_{(d+1):D}^{\mathrm{Jac},x} \cdot \left\| f_{\widetilde{V}_d,\widetilde{U}_d}\left(f_{\widetilde{V}_{d-1},\widetilde{U}_{d-1}}(\cdots)\right) - f_{\widetilde{V}_d,U_d}\left(f_{\widetilde{V}_{d-1},\widetilde{U}_{d-1}}(\cdots)\right) \right\|_2
$$

$$
+ \sum_{d=1}^{D} B_{(d+1):D}^{\mathrm{Jac},x} \cdot \left\| f_{\widetilde{V}_d,U_d}\left(f_{\widetilde{V}_{d-1},\widetilde{U}_{d-1}}(\cdots)\right) - f_{V_d,U_d}\left(f_{\widetilde{V}_{d-1},\widetilde{U}_{d-1}}(\cdots)\right) \right\|_2
$$

$$
\overset{(i)}{\leq} \sum_{d=1}^{D} B_{(d+1):D}^{\mathrm{Jac},x} \cdot \left\| \widetilde{V}_d \sigma\left(\widetilde{U}_d \cdot f_{\widetilde{V}_{d-1},\widetilde{U}_{d-1}}(\cdots)\right) - \widetilde{V}_d \sigma\left(U_d \cdot f_{\widetilde{V}_{d-1},\widetilde{U}_{d-1}}(\cdots)\right) \right\|_2
$$

$$
+ \sum_{d=1}^{D} B_{(d+1):D}^{\mathrm{Jac},x} \cdot \left\| \widetilde{V}_d \sigma\left(U_d \cdot f_{\widetilde{V}_{d-1},\widetilde{U}_{d-1}}(\cdots)\right) - V_d \sigma\left(U_d \cdot f_{\widetilde{V}_{d-1},\widetilde{U}_{d-1}}(\cdots)\right) \right\|_2
$$

$$
\overset{(ii)}{\leq} \sum_{d=1}^{D} B_{(d+1):D}^{\mathrm{Jac},x} \cdot \left( \left\| U_d - \widetilde{U}_d \right\|_2 \|V_d\|_2 + \left\| V_d - \widetilde{V}_d \right\|_2 \|U_d\|_2 \right) \left\| f_{\widetilde{V}_{d-1},\widetilde{U}_{d-1}}(\cdots) \right\|_2, \tag{21}
$$

where $(i)$ and $(ii)$ from the entry-wise 1–Lipschitz continuity of $\sigma(\cdot)$. In addition, for any $d \in [D]$, we further have

$$
\left\| f_{\widetilde{V}_{d-1},\widetilde{U}_{d-1}}\left(\cdots f_{\widetilde{V}_1,\widetilde{U}_1}(x)\right) \right\|_2 = \|J_{1:d}^{x} \cdot x\|_2 \leq B_{1:(d-1)}^{\mathrm{Jac},x} \cdot \|x\|_2. \tag{22}
$$

Combining (21) and (22), we obtain

$$
\left\| f_{V_D,U_D}\left(\cdots f_{V_1,U_1}(x)\right) - f_{\widetilde{V}_D,\widetilde{U}_D}\left(\cdots f_{\widetilde{V}_1,\widetilde{U}_1}(x)\right) \right\|_2
$$

$$
\leq \sum_{d=1}^{D} B_{(d+1):D}^{\mathrm{Jac},x} \cdot B_{1:(d-1)}^{\mathrm{Jac},x} \cdot \|x\|_2 \cdot \left( \left\| V_d - \widetilde{V}_d \right\|_{\mathrm{F}} \cdot \|U_d\|_2 + \left\| U_d - \widetilde{U}_d \right\|_{\mathrm{F}} \cdot \left\| \widetilde{V}_d \right\|_2 \right)
$$

$$
\leq B_{\backslash d,2}^{\mathrm{Jac},x} \max_d \left(B_{V_d,2} + B_{U_d,2}\right) R \sum_{d=1}^{D} \cdot \left( \left\| V_d - \widetilde{V}_d \right\|_{\mathrm{F}} + \left\| U_d - \widetilde{U}_d \right\|_{\mathrm{F}} \right)
$$

$$
\leq B_{\backslash d,2}^{\mathrm{Jac},x} \max_d \left(B_{V_d,2} + B_{U_d,2}\right) R\sqrt{2D} \cdot \sqrt{\sum_{d=1}^{D} \left\| V_D - \widetilde{V}_D \right\|_{\mathrm{F}}^2 + \sum_{d=1}^{D} \left\| U_D - \widetilde{U}_D \right\|_{\mathrm{F}}^2}.
$$

$\square$

On the other hand, for any $d \in [D]$, we have

$$
\begin{aligned}
&\left\| f_{V_d, U_d} \left( \cdots f_{V_1, U_1}(x) \right) \right\|_2 \\
&\overset{(i)}{\leq} \|V_d\|_2 \|U_d\|_2 \cdot \left\| f_{V_{d-1}, U_{d-1}} \left( \cdots f_{V_1, U_1}(x) \right) \right\|_2 + \left\| f_{V_{d-1}, U_{d-1}} \left( \cdots f_{V_1, U_1}(x) \right) \right\|_2 \\
&\overset{(ii)}{\leq} \prod_{i=1}^{d} \left( \|V_i\|_2 \|U_i\|_2 + 1 \right) \cdot \|x\|_2 .
\end{aligned}
\tag{23}
$$

where $(i)$ is from the entry-wise 1–Lipschitz continuity of $\sigma(\cdot)$ and $(ii)$ is from recursively applying the same argument.

Let $p_1 = \cdots = p_D = p$ and $q_1 = \cdots = q_D = q$. Then following the same argument as in the proof of Theorem 1 and (23), we have

$$
\begin{aligned}
\alpha &= \sup_{f \in \mathcal{F}_{D, \|\cdot\|_2}, x \in \mathcal{X}_m} g_\gamma \left( f\left(\mathcal{V}_D, \mathcal{V}_D, x\right) \right) \leq \frac{R \cdot \prod_{d=1}^{D} \left( B_{V_d,2} B_{U_d,2} + 1 \right)}{\gamma}, \\
L_w &\leq \max_{x \in \mathcal{X}_m} B_{\backslash d,2}^{\mathrm{Jac},x} \max_d \left( B_{V_d,2} + B_{U_d,2} \right) R \sqrt{2D}, \\
K &= \sqrt{\sum_{d=1}^{D} \|V_d\|_F^2 + \|U_d\|_F^2} \leq \sqrt{pD} \cdot \max_d \left( B_{V_d,2} + B_{U_d,2} \right), \text{ and } h = 2Dpq,
\end{aligned}
$$

Combining Lemma 3, and Lemma 4, Lemma 5, we have

$$
\begin{aligned}
\mathcal{R}_m(\mathcal{G}) &\lesssim \frac{\alpha \sqrt{h \log \frac{K L_w \sqrt{m}}{\alpha \sqrt{h}}}}{\sqrt{m}} \\
&\leq \frac{R \cdot \prod_{d=1}^{D} \left( B_{V_d,2} B_{U_d,2} + 1 \right) \cdot \sqrt{Dpq \cdot \log \left( \frac{B_{\backslash d,2}^{\mathrm{Jac}} \max_d \left( B_{V_d,2} + B_{U_d,2} \right) R \sqrt{m/q}}{\sup_{f \in \mathcal{F}_{D, \|\cdot\|_2}, x \in \mathcal{X}_m} g_\gamma (f(\mathcal{V}_D, \mathcal{V}_D, x))} \right)}}{\gamma \sqrt{m}} .
\end{aligned}
$$

# F  SPECTRAL BOUND FOR $W_d$ IN CNNs WITH MATRIX FILTERS

We provide further discussion on the upper bound of the spectral norm for the weight matrix $W_d$ in CNNs with matrix filters. In particular, by denoting $W_d$ using submatrices as in (7), i.e.,

$$
W_d = \left[ W_d^{(1)\top} \cdots W_d^{(n_d)\top} \right]^\top \in \mathbb{R}^{p_d \times p_{d-1}},
$$

we have that each block matrix $W_d^{(j)}$ is of the form

$$
W_d^{(j)} = \begin{bmatrix}
W_d^{(j)}(1,1) & W_d^{(j)}(1,2) & \cdots & W_d^{(j)}\left(1, \sqrt{p_{d-1}}\right) \\
W_d^{(j)}(2,1) & W_d^{(j)}(2,2) & \cdots & W_d^{(j)}\left(2, \sqrt{p_{d-1}}\right) \\
\vdots & \vdots & \ddots & \vdots \\
W_d^{(j)}\left(\frac{\sqrt{p_{d-1}k_d}}{s_d}, 1\right) & W_d^{(j)}\left(\frac{\sqrt{p_{d-1}k_d}}{s_d}, 2\right) & \cdots & W_d^{(j)}\left(\frac{\sqrt{p_{d-1}k_d}}{s_d}, \sqrt{p_{d-1}}\right)
\end{bmatrix},
\tag{24}
$$

where $W_d^{(j)}(i,l) \in \mathbb{R}^{\frac{\sqrt{p_{d-1}k_d}}{s_d} \times \sqrt{p_{d-1}}}$ for all $i \in \left[\frac{\sqrt{p_{d-1}k_d}}{s_d}\right]$ and $l \in \left[\sqrt{p_{d-1}}\right]$. Particularly, off-diagonal blocks are zero matrices, i.e., $W_d^{(j)}(i,l) = 0$ for $i \neq l$. For diagonal blocks, we have

$$W_d^{(j)}(i,i) = \begin{bmatrix} \underbrace{w^{(j,1)}}_{\in \mathbb{R}^{\sqrt{k_d}}} \underbrace{0 \cdots\cdots\cdots 0}_{\in \mathbb{R}^{\sqrt{\frac{p_{d-1}}{k_d}} - \sqrt{k_d}}} \cdots\cdots\cdots\cdots \underbrace{w^{(j,\sqrt{k_d})}}_{\in \mathbb{R}^{\sqrt{k_d}}} \underbrace{0 \cdots\cdots\cdots\cdots\cdots 0}_{\in \mathbb{R}^{\sqrt{\frac{p_{d-1}}{k_d}} - \sqrt{k_d}}} \\ \underbrace{0 \cdots 0}_{\in \mathbb{R}^{\frac{s_d}{\sqrt{k_d}}}} \underbrace{w^{(j,1)}}_{\in \mathbb{R}^{\sqrt{k_d}}} \underbrace{0 \cdots\cdots\cdots\cdots 0}_{\in \mathbb{R}^{\sqrt{\frac{p_{d-1}}{k_d}} - \sqrt{k_d}}} \cdots\cdots\cdots\cdots\cdots \underbrace{w^{(j,\sqrt{k_d})}}_{\in \mathbb{R}^{\sqrt{k_d}}} \underbrace{0 \cdots\cdots\cdots\cdots 0}_{\in \mathbb{R}^{\sqrt{\frac{p_{d-1}}{k_d}} - \sqrt{k_d} - \frac{s_d}{\sqrt{k_d}}}} \\ \vdots \\ \underbrace{w^{(j,1)}_{\left\{\frac{s_d}{\sqrt{k_d}}\right\}}} \underbrace{0 \cdots\cdots\cdots 0}_{\in \mathbb{R}^{\sqrt{\frac{p_{d-1}}{k_d}} - \sqrt{k_d}}} \cdots\cdots \underbrace{w^{(j,\sqrt{k_d})}}_{\in \mathbb{R}^{\sqrt{k_d}}} \underbrace{0 \cdots\cdots\cdots\cdots\cdots 0}_{\in \mathbb{R}^{\sqrt{\frac{p_{d-1}}{k_d}} - \sqrt{k_d}}} w^{(j,1)}_{\left\{\frac{s_d}{1}\right\}} \end{bmatrix} . \tag{25}$$

where $w^{(j,1)}_{\left\{\frac{s_d}{1}\right\}} = w^{(j,1)}_{1:\frac{s_d}{\sqrt{k_d}}} \in \mathbb{R}^{\frac{s_d}{\sqrt{k_d}}}$ and $w^{(j,1)}_{\left\{\frac{s_d}{\sqrt{k_d}}\right\}} = w^{(j,1)}_{\left(\sqrt{k_d} - \frac{s_d}{\sqrt{k_d}} + 1\right):\sqrt{k_d}} \in \mathbb{R}^{\frac{s_d}{\sqrt{k_d}}}$. Combining (24) and (25), we have that the stride for $W_d^{(j)}$ is $\frac{s_d^2}{k_d}$. Using the same analysis for Corollary 2. We have $\|W_d\|_2 = 1$ if $\sqrt{\sum_i \|w^{(j,i)}\|_2^2} = \frac{k_d}{s_d}$.

For image inputs, we need an even smaller matrix $W_d^{(j)}(i,i)$ with fewer rows than (25), denoted as

$$W_d^{(j)}(i,i) = \begin{bmatrix} \underbrace{w^{(j,1)}}_{\in \mathbb{R}^{\sqrt{k_d}}} \underbrace{0 \cdots\cdots\cdots 0}_{\in \mathbb{R}^{\sqrt{\frac{p_{d-1}}{k_d}} - \sqrt{k_d}}} \cdots\cdots\cdots\cdots \underbrace{w^{(j,\sqrt{k_d})}}_{\in \mathbb{R}^{\sqrt{k_d}}} \underbrace{0 \cdots\cdots\cdots\cdots\cdots 0}_{\in \mathbb{R}^{\sqrt{\frac{p_{d-1}}{k_d}} - \sqrt{k_d}}} \\ \underbrace{0 \cdots 0}_{\in \mathbb{R}^{\frac{s_d}{\sqrt{k_d}}}} \underbrace{w^{(j,1)}}_{\in \mathbb{R}^{\sqrt{k_d}}} \underbrace{0 \cdots\cdots\cdots\cdots 0}_{\in \mathbb{R}^{\sqrt{\frac{p_{d-1}}{k_d}} - \sqrt{k_d}}} \cdots\cdots\cdots\cdots\cdots \underbrace{w^{(j,\sqrt{k_d})}}_{\in \mathbb{R}^{\sqrt{k_d}}} \underbrace{0 \cdots\cdots\cdots\cdots 0}_{\in \mathbb{R}^{\sqrt{\frac{p_{d-1}}{k_d}} - \sqrt{k_d} - \frac{s_d}{\sqrt{k_d}}}} \\ \vdots \\ \underbrace{0 \cdots\cdots\cdots\cdots\cdots 0}_{\in \mathbb{R}^{\sqrt{\frac{p_{d-1}}{k_d}} - \sqrt{k_d}}} \underbrace{w^{(j,1)}}_{\in \mathbb{R}^{\sqrt{k_d}}} \underbrace{0 \cdots\cdots\cdots\cdots 0}_{\in \mathbb{R}^{\sqrt{\frac{p_{d-1}}{k_d}} - \sqrt{k_d}}} \cdots\cdots\cdots\cdots \underbrace{w^{(j,\sqrt{k_d})}}_{\in \mathbb{R}^{\sqrt{k_d}}} \end{bmatrix} . \tag{26}$$

Then $\|W_d\|_2 \leq 1$ still holds if $\sqrt{\sum_i \|w^{(j,i)}\|_2^2} = \frac{k_d}{s_d}$ since $W_d$ generated using (26) is a submatrix of $W_d$ generated using (25).

