# OpenReview forum: "On Tighter Generalization Bounds for Deep Neural Networks: CNNs, ResNets, and Beyond"
_ICLR.cc/2019/Conference_

### Official Review · AnonReviewer3 · 2018-10-30

**Rating:** 7
**Confidence:** 4

**Review:**

After rebuttal: The authors have nicely addressed my comments. I have increased my rating to 7.
~~~~~~~~~~~~~~~~~~~~~~~~~~~~~~~~~~~~~~~~~~~~~~~~~~~~~~~~~~~~~~~~~~~~~~~~~
This paper proposes a generalization error bound for DNNs (and their generalizations) based on the depth and width of the networks, as well as the spectral norm of weight matrices. The proposed work is shown to provide a tighter generalization error bounds compared with a few existing literatures.

Pros: This paper makes theoretical contributions to the understanding of DNNs. This is an important but difficult task. As a theoretical paper, this one is relatively easy to follow.

Cons: In spite of its theoretical contributions, this paper has a few major issues.

Q1: This paper fails to fairly compare with the most recent work, Arora et al. (2018), Zhou and Feng (2018). For instance, Arora et al. (2018) uses error-resilience parameters instead of the norms of weight matrices to obtain a better generalization error. The authors claim that the error-resilience parameters are less interpretable than the norms of weight matrices. This claim could be subjective and is not convincing.

Q2: The error bounds of Bartlett et al. (2017), Neyshabur et al. (2017) could be improved for low-rank weight matrices, in which case the proposed Theorem 1 is tighter only if $p \le D^2$. This holds only when DNN is very deep. Can theorem 1 be improved by similarly considering the low-rankness of weight matrices?

Q3: In Corollary 2, the error bound for CNN, the authors assume that the filters are orthogonal with unit norm. Can the authors provide some justification on the orthogonal filters? In addition, Zhou and Feng (2018) have achieved similar bound for CNN. Can the authors provide some justification why this latest result is not included in Table 2?

---

> ### Author Response · Authors · 2018-11-20
> **Extended discussion, numerical results, and refined analysis**
>
> We thank the reviewer for the careful reading and recognizing our contributions.
>
> 1. We have added some discussion to compare with the [1,2] after corollary 1 in the bounded loss case. Moreover, we extended the numerical result to compare with [1,2] in Appendix A.1 for the trained network on CIFAR10. Please refer further details therein. Our results show that our bound is tighter than the existing works based on the bounded function [1,2].
>
> 2. Thanks to the reviewer for pointing this out, we updated our analysis to incorporate the rank of weight matrices in the revision. In the updated results, we have sqrt(pr) (r is the rank) dependence, which is of the same order with [3,4] in terms of the width. To this end, our bound is strictly better than [3,4] by a margin at least D in all scenarios of ranks.
>
> 3. The orthogonal and normalized filters in CNNs have shown comparable or even improved empirical results than their non-orthogonal counterpart [5,6]. This motivates us to analyze this setting for CNNs, which also provide a way to quantify the spectral of weight matrices for CNNs tightly based on the filter size and stride size. We further clarified this in the revision. In addition, we provided further discussion to compare with [2] in Section 3.2 and extended numerical evaluations in Appendix A.1 in the bounded loss case. This support that our new bound is tighter than [2] in this case. Also, note that the orthogonality is not a must as we discussed in the numerical evaluation (Section 4.5). When the filters are not orthogonal, our bound reduces to the number of total parameters in filters (# filters * filter size). This is one of the benefits of our analytic pipeline that allows obtaining tighter bounds based on the number of parameters rather than the width of weight matrices.
>
> [1] Arora et al. Stronger generalization bounds for deep nets via a compression approach.
> [2] Zhou and Feng. Understanding generalization and optimization performance of deep cnns.
> [3] Bartlett et al. Spectrally-normalized margin bounds for neural networks.
> [4] Neyshabur et al. A pac-bayesian approach to spectrally-normalized margin bounds for neural networks.
> [5] Huang et al. Orthogonal weight normalization: Solution to optimization over multiple dependent stiefel manifolds in deep neural networks.
> [6] Xie et al. All you need is beyond a good init: Exploring better solution for training extremely deep convolutional neural networks with orthonormality and modulation.

---

### Official Review · AnonReviewer2 · 2018-10-31
**An improved and new characterization of generalization bound**

**Rating:** 7
**Confidence:** 4

**Review:**

The rebutal and the revision of the paper solve my comments.
~~~~~~~~~~~~~~~~~~~~~~~~~~~~~~~~~~~~
The paper presents a new characterization of generalization error bound for general deep neural networks in terms of the depth and width of the networks and the spectral norm of weight matrices. The proof follows the setting of Bartlett et al. 2017 with new development on the Lipschitz properties of neural networks.

Pros:
1. The paper provides a solid improvement over previous bounds on generalization error.
2. The presentation of the result and proofs is clear and easy to follow.
3. It does case studies specially on widely used network structures CNN, ResNet, etc.

Con:
1. It is not clear whether the bound is vacuous or not, as discussed in Arora et al. 2018.  If it is vacuous, it is hard to justify the claims that given the generalization error, which is vacuous for all bounds, the paper's bound allows the choices of larger dimensions of parameters and larger spectral norms of weight matrices.
2.  The L_w has the factor "products of B_{d,2}s" which, however, does not show up in the final generalization bound (The equation right above Appendix B). This products may introduce an additional $D$ under sqrt changing D to D^2 under the sqrt, which changes the order. The authors should give some explanation on this.

3. Is the assumption on the orthogonal and normalized filters in CNN a must thing for the argument or just for convenience of the presentation? The paper should be clearer about this point.

4. The RHS of the equation  in Lemma 2 misses terms related with B_{d,2}.
5. Typos: Find one typo in Page 3  "deﬁed as"

---

> ### Author Response · Authors · 2018-11-20
> **Extended discussion, experiments, and refined analysis**
>
> We thank the reviewer for recognizing our contributions.
>
> 1. As in existing norm based bounds, it can be vacuous (e.g., when the norms of weight matrices are large) if we do not have further structural conditions on the networks. Our effort here is to tighten the potentially vacuous bound from two directions: (1) reduce the dependence in terms of the depth and width; (2) reduce the product of norms to the loss function output when it is bounded. With these improvements, we can obtain non-vacuous bounds. For example, in the case of CNNs with a bounded loss, our generalization bound can be << 1 given a moderate training sample size. An extended numerical result is provided in Appendix A.1 to support this claim.
>
> 2. We updated the proof of Theorem 1 in terms of the L_w using the spectral norm of Jacobian operators instead of the product of spectral norms of weight matrices. This is a tighter result since the spectral norm of Jacobian is significantly smaller than the product of spectral norms of weight matrices. For example, when we constrain the network to be obtained from stochastic gradient descent using randomly initialized weights, the spectral norm of Jacobian is significantly smaller than the product of spectral norms of weight matrices that can be exponential on depth in general. To illustrate this, we provided an extended numerical result on the empirical distributions of the norm of Jacobian and the product of spectral norms in Appendix A.2 and A.3. We can observe that the values of the norm of Jacobian are orders smaller than the values product of norms, and the former quantity increases significantly slower than the latter when the depth increases (even slower than some low degree polynomial on the depth). When we consider the network functions obtained using proper training procedures, we do have sqrt(D) dependence in ERC rather than D.
>
> 3. The orthogonal and normalized filters in CNNs have shown comparable or even improved empirical results than their non-orthogonal counterpart [1,2]. This motivates us to analyze this setting for CNNs. In addition, the orthogonal filters also provide a way to quantify the spectral of weight matrices for CNNs tightly based on the filter size and stride size. We further clarified this in the revision. On the other hand, the orthogonality is not a must as we discussed in the numerical evaluation (Section 4.5). When the filters are not orthogonal, our bound reduces to the number of total parameters in filters (# filters * filter size). This is one of the benefits of our analytic pipeline that allows obtaining potentially tighter bounds based on the number of parameters rather than the width of weight matrices.
>
> 4 and 5. The typos are corrected.
>
> [1] Huang et al. Orthogonal weight normalization: Solution to optimization over multiple dependent stiefel manifolds in deep neural networks.
> [2] Xie et al. All you need is beyond a good init: Exploring better solution for training extremely deep convolutional neural networks with orthonormality and modulation.

---

### Official Review · AnonReviewer1 · 2018-11-03
**Different Take on Generalization - Size Dependent Bounds**

**Rating:** 5
**Confidence:** 3

**Review:**

The paper provides a generalization bound for multi-layered deep neural networks in terms of dimensions rather than norms. The bound is derived by controlling Rademacher complexity of the Ramp loss under the Lipschitzness of the network as a parametric function in Depth * Width ^2 number of parameters, and then using standard L-2 covering and Dudley Integral. They extend this technique for CNNs, Resnets, Hyper-spherical Networks, etc and provide specialized bounds for each case. In the end, the authors provide comparisons to the existing bounds.

Although intended, the bound in Theorem-1 depends on the number of parameters and hold only if m > d * (p^2) = number of parameters (from the last line of proof of Lemma 3, we need \beta < \alpha and thus m > h). Such bounds are already know in the literature (see Anthony and Bartlett, 1999). Adaptive (completely norm dependent, like Bartlett et. al. 2017) bounds will be better than explicit dimension dependent bounds. The comparison in Figure-1 which suggest their bound to be better is unfair because they are comparing their specialized bounds for CNN to generic bounds for standard feedforward networks. Same for comparison in Table-2.

It was already established in Theorem 3.4 (Bartlett et al. 2017) that spectral norms are necessary for any generalization bounds for Deep Neural Networks, thus voiding the claims made in the paper (and discussion) about the importance of spectral norms.

Typos / Errors :
1. Statement of Lemma 2 does not contain the spectral norms terms.
2. The third equation in Page 13 should be K <= \sqrt{pD} max B_{d, 2}; and this changes the bound further.

The paper introduces some new techniques on mathematical analysis of specialized neural networks.

---

> ### Author Response · Authors · 2018-11-20
> **Refined analysis and extended numerical results**
>
> We thank the reviewer for careful reading and helpful comments. We do have the same dependence with the VC type of bound in terms of the number of parameters, but we are considering essentially a smaller class of network functions (than that considered in the VC type of bound) with norm constraints. In the case when all norms are bounded (by 1) or the loss function is bounded (e.g., the ramp loss < 1), then our resulting bound can be smaller than the VC type of bound.
>
> We want to remark that we have updated our analysis for DNNs in terms of the rank of weight matrices, which may lead to tighter results when the weight matrices are low-rank, analogous to [1,2]. Our result for CNNs is a direct derivation from Theorem 1 (for DNNs). This is one benefit of our analysis based on the Lipschitz property on the parameters, which allows us to directly reduce from pr (r is the rank of W) to k^2 in the CNNs case. For the other comparing results in Figure 1, to the best of our knowledge, there is no such direct simplification from the general case to the CNNs case from their analysis. In other words, they have the same order of bounds for DNNs and CNNs. Note that the ranks are of the same order with width p in CNNs when the filters are linear independent, and we already simplified the norms in the other comparing results as in our result for CNNs in Table 2.
>
> Moreover, we want to remark that Thm 3.4 in [1] is the lower bound of the ERC for the neural network function, which did not take the loss function into consideration. In other words, it does not conflict our case when the loss is bounded, which can lead to tighter dependence in terms of the loss function bound rather than the product of norm. Note that our result shown in Figure 1 is the worse case of Corollary 1. In practice, the loss function for a trained network has a significantly smaller scale than \prod B_{d,2}, which mean our result in Figure 1 is in fact significantly smaller than what is shown. This is another benefit of our analysis that allows us to only depend on the output of loss function, rather than the product of norms that the comparing results in Figure 1 cannot avoid due to the nature of their analyses. We have added an extended experiment in Appendix A.1 for the bounded loss case. One can find that the generalization bound in the bounded loss case can be significantly smaller than the norm based result. We also added a discussion in Section 3.2 and Appendix A.1 to compare with existing output based bounds [3,4] and show that our output based result is tighter.
>
> In addition, from our observation on the trained network using real data, the dependence on the network sizes in other norm based results (e.g., the terms in Bound 1 and 2 in Figure 1 excluding the term of the product of norms) are significantly larger than the number of parameters. Based on the updated Theorem 1, [1,2] are always larger than our bound by a margin at least D, including the low-rank cases.
>
> The typos are corrected.
>
> [1] Bartlett et al. Spectrally-normalized margin bounds for neural networks.
> [2] Neyshabur et al. A pac-bayesian approach to spectrally-normalized margin bounds for neural networks.
> [3] Arora et al. Stronger generalization bounds for deep nets via a compression approach.
> [4] Zhou and Feng. Understanding generalization and optimization performance of deep cnns.

---

### Public Comment · (anonymous) · 2018-11-16
**Why is the dependence on parameters in Corollary 1 \sqrt{Dp^2} and not \sqrt{D^2 p^2}?**

I've a couple of questions about the proof of Lemma 3 and Corollary 1 and I'll be glad if the authors can help clarify.

First, at the end of page 12, there's an upper bound of the form  alpha log (1/alpha).  Equation 15 upper bounds alpha. How do you use this upper bound on alpha to upper bound the 1/alpha within the log term later? Perhaps I'm missing something here, is it possible to directly plug in the upper bound on alpha in alpha log(1/alpha)?

Second, in the proof of corollary 2, assuming we can directly plug in the tighter upper bound on alpha, we will have a log (L_w/ alpha) term where L_w scales with the product of spectral norms (which wouldn't get cancelled now because alpha is potentially much smaller than that). Wouldn't this result in an extra dependence on D as sqrt{D*log max_d B_{d,2}}?

---

> ### Author Response · Authors · 2018-11-20
> **Refined analysis**
>
> We updated the proof of Theorem 1 in terms of the L_w using the spectral norm of Jacobian operators instead of the product of spectral norms of weight matrices. This is a tighter result since the spectral norm of Jacobian is significantly smaller than the product of spectral norms of weight matrices. For example, when we constrain the network to be obtained from stochastic gradient descent using randomly initialized weights, the spectral norm of Jacobian is significantly smaller than the product of spectral norms of weight matrices that can be exponential on depth in general. To illustrate this, we provided an extended numerical result on the empirical distributions of the norm of Jacobian and the product of spectral norms in Appendix A.2. We can observe that the values of the norm of Jacobian are approximately 2 orders smaller than the values product of norms on a real dataset. When we consider the network functions obtained using proper training procedures, we do have sqrt(D) dependence in ERC rather than D. The same argument applies to Corollary 2 for CNNs. We also provided details of quantities alpha, L_w, and K in all cases for completeness.

---

> > ### Public Comment · (anonymous) · 2018-11-21
> > **Response**
> >
> > Hi, Thanks a lot for responding to my comment :) Thank you for adding more details about the quantities for completeness.
> >
> > My question still remains, so let me try rephrase (and I have an additional question with the introduction of the Jacobian terms).
> >
> > 1. About my first question in the previous comment: As for the alpha in the denominator of log(1/\alpha), if I understand it right, it seems like you have changed the results so that the expression is not simplified anymore and is instead abstracted away as a term C_1. I'm not sure I understand how to parse this result or upper bound this term in terms of D.
> >
> > 2. About my second question in the previous comment:  Now in corollary 2, you have log(B^jac / alpha) where alpha is at best a constant. So the term would scale as log(B^jac) and your claim is that B^jac is significantly smaller than the product of spectral norms (the log of which will surely grow linearly with D). I completely agree that this new term is way smaller in magnitude -- but when we take a logarithm, what exactly is the dependence on the depth? I'm not sure how this claim holds:
> >
> > "Thus, logC1 can be considered as a constant that is (almost) in- dependent of D"
> >
> > My suspicion is that the log could potential linearly grow with depth, perhaps with a smaller proportionality constant. This could be verified through experiments where the depth is varied. Currently, the discussion below Corollary 2 (and the second column in Table 2) assumes that this new log term is independent of depth without experimental justification. My suspicion about this unaccounted-for depth dependence holds good for Theorem 1 too, but I'm not sure how to study that term because I don't know how to understand the dependence of C_1 on D there, given that its denominator is not simplified.
> >
> > 3. Finally, and most importantly, the new class of functions in equation 1, have a bounded Jacobian spectral norm. Like you've correctly noted, the spectral norm of the Jacobian is not purely a property of the weights but depends on the input datapoints. Thus, you've effectively assumed this bound holds on all inputs, including not just training inputs but also test inputs. This seems like a strong assumption (especially from the point of view of a generalization guarantee), which I feel the paper could be more transparent about in the introduction and the theorem statements. This comment of course doesn't apply to your previous results which effectively bounded the Jacobian spectral norms in terms of the product of spectral norms, a quantity independent of the input.

---

> > > ### Author Response · Authors · 2018-11-23
> > > **Further responses**
> > >
> > > 1. Since alpha = sup_{f,x} g(f(x)) and we consider spectrally bounded weight, for Lipschitz loss g, this sup is equal to the product of spectral norm (when the direction of largest singular values of all weight matrices perfectly align with certain data). In this case, the value B^Jac is also the product of spectral norms, which allow the numerator and denominator in the log factor to cancel out each other, and there is no exponential dependence of D in C_1. This is fundamentally different with existing results, e.g., [1], where they have product of Frobenius norms divided by the max output norm, which even in the perfectly aligned case above (so that the output norm can take the sup value) still has linear dependence of D in the log factor since Frobenius norm product divided by the spectral norm product is of order rank^(D/2).
> > >
> > > 2. For the bounded loss, i.e., alpha is a constant, as we discussed in the revision, B^Jac is of the order of spectral norm product in the worst case. However, what we (and most practitioners) care about is not the worst-case scenarios. Instead, when the networks are trained using sgd type of algorithms with random initializations, our preliminary numerical results show that the norm of B^Jac increase at a significantly slower rate than exponential on D (even slower than some low degree of polynomial on D) when the depth increases. An accurate quantification of the rate of B^Jac in terms of the depth for networks of interest is a desirable future effort (yet beyond the scope of this paper).
> > >
> > > 3. It is true that the jacobian depends on not only the weight matrices but also the input data. But in practice, the weight matrices of interested (trained via sgd and random initialization on reasonable datasets) also depend on the input data in an implicit way. For example, the weight matrices obtained using training data with random labels are not interesting even though they have zero error on training data, since they will not generalize to test data. Therefore, analyzing the generalization bound by incorporating the optimization procedure with good generalization performance may potentially lead to more powerful results than considering the global parameter space (where the sup takes the worst case), which is not well understood yet and a potential future direction. In the current paper, we simply assume the norm of jacobian to be bounded as a constant without further ado.
> > >
> > > [1] Golowich et al. Size-Independent Sample Complexity of Neural Networks.

---

> > > > ### Public Comment · (anonymous) · 2018-11-23
> > > > **My concerns about the refined analysis**
> > > >
> > > > Hi, Thanks for responding. I will skip point 1 because my disagreement with the other points are stronger and more conceptual; my disagreement on point 1 is more arithmetic and doesn't bother me as much as 2 and 3.
> > > >
> > > > About 2), thanks for running the preliminary experiments in a short period of time. I think the paper requires plots that show the dependence of B^jac on the depth to justify the comment, "Thus, logC1 can be considered as a constant that is (almost) in- dependent of D"
> > > >  and the absence of a \sqrt{D} in all your results. It'd be great if at the least your preliminary results are added to the paper.  If I understand the intention of the new refined analysis in the paper it is that, the old results in Corollary did have an extra \sqrt{D} dependence (that was unaccounted for) because of the log(product of spectral norm)-dependence, and the Jacobian term hopefully get rids of this dependence.  If the authors believe that an accurate quantification of the rate of B^Jac in terms of the depth for networks of interest is beyond the scope of this paper, then the above quoted line and the dependence of depth expressed in the results become unjustified within the context of this paper, in my opinion. I must emphasize that I'm not questioning the numerical value of the bounds, but the dependence on depth that is claimed in the introduction and in the discussions after the theorem statements.
> > > >
> > > > About 3) I agree that identifying hypotheses classes that better reflect what is learned by SGD is important to understand generalization. But I completely disagree with your reasoning that
> > > > "But in practice, the weight matrices of interested (trained via sgd and random initialization on reasonable datasets) also depend on the input data in an implicit way. "
> > > >
> > > > First of all the weight matrices don't depend on arbitrary inputs including unseen inputs -- they depend only on training data. Furthermore, given these weight matrices, the norms of these weight matrices can be computed independent of any input. On the other hand, the norms of the Jacobians of these matrices depend on the actual input used during test time.
> > > >
> > > > In summary, given the learned weight matrices, a) the norms of these matrices are independent of any input but b) the norms of the Jacobians of these matrices are dependent on the input. This makes a lot of difference.
> > > >
> > > > Specifically, the assumed bound on the Jacobians for all inputs potentially assumes "a part of the generalization", and therefore your resulting bound potentially explains only the remaining part of the generalization. I want to emphasize that I am not at all questioning the validity of your final bound or the proofs, but I suggest that, since this is a strong assumption, the paper should be more transparent about it in the introduction and in the theorem statements.

---

> > > > > ### Author Response · Authors · 2018-11-24
> > > > > **Further response**
> > > > >
> > > > > About 2. We will include the numerical result for scales of norms of the Jacobian term once we have it summarized.
> > > > >
> > > > > About3. I think there is some misunderstanding here. The empirical Rademacher complexity only depends on the training data, which means the Jacobian term for upper bounding the empirical Rademacher complexity only depends on the training data.
> > > > >
> > > > > On the other hand, I do not think the independence of the norms of weight matrices on training data after the training process is good for understanding/interpreting the generalization performance. As in our example mentioned in the previous response, we may train the network using a set of data containing (partially) perturbed labels with very small training error, but these learned weights are not good ones to generalize even their norms do not depend on the training data anymore after the learning procedure. In other words, I think both the optimization procedure and data properties are important to understand the generalization of DNNs and obtain good generalization performance, but we do not have a clear way yet to characterize them.
> > > > >
> > > > > In addition, the Jacobian is already a relatively easy-to-interpret and mild term for the network function in understanding the generalization. Some recent results on improved generalization bound also contain parameters that require post-calculation depending on training data. For example, [1] needs to calculate jacobian and layer cushion parameters (these only depends on the training data with the reasoning above). Note that [1] obtained better error bounds (see their Fig. 4) essentially because they do not depend on the norm of weight, but on the max output of network functions. This agrees with our idea in essence that a better generalization bound may be data dependent. Just like we can constrain the norm of the weight matrices in the learning process to obtain uniformly bounded norms on weight matrices, we may also do so to add norm constraints on the Jacobian terms to obtain bounded norms on jacobian (that is independent on D). It will be interesting to develop a computationally efficient algorithm for this purpose in the future study (yet the simple exact spectral norm constraint on weight matrices alone is still computationally challenging).
> > > > >
> > > > > [1] Arora et al. Stronger generalization bounds for deep nets via a compression approach.

---

> > > > > > ### Public Comment · (anonymous) · 2018-11-24
> > > > > > **ERC implicitly depends on multiple draws of the dataset**
> > > > > >
> > > > > > Thanks for having a patient & detailed discussion about this.
> > > > > >
> > > > > > >>>>> The empirical Rademacher complexity only depends on the training data, which means the Jacobian term for upper bounding the empirical Rademacher complexity only depends on the training data.
> > > > > >
> > > > > > I believe there's a subtle point where this claim should run into trouble: ERC explicitly depends on a specific draw of the dataset, but also implicitly depends on other draws as well!
> > > > > >
> > > > > > On a dataset S, let h_S be the function learned by SGD.  Within the sup of the ERC, the summation over the Rademacher variables might be over a particular S. However, the supremum is over a class of hypothesis that corresponds to the hypotheses learned by the algorithm across many other draws of a training dataset S' i.e., sup_{h(S') for many independent draws of S'}. If you want your ERC to simplify in terms of the Jacobian bounds, then you are implicitly assuming that the weights h_{S'} learned on some other draw of the dataset S' have Jacobian norms that are bounded on the dataset S. In other words, without assuming that the Jacobian norms of your functions h_{S'} are bounded on all inputs, including training and unseen inputs for that function (S' and S, respectively), you cannot simplify your ERC in terms of the Jacobian.
> > > > > >
> > > > > > As a side note, Equation (1) doesn't have a superscript on J; it's not clear if the bounded norm assumption that is currently stated in the paper is only on training data.
> > > > > >
> > > > > > About the reference to [1]: [1] indeed make such assumptions only on the training data. But their generalization bound holds for a compressed network, which seems like the price they pay for not making such assumptions on test data. If I understand their paper right, if they could make such assumptions on the test data, they should be able to present a bound on the uncompressed network too (although that would be a strong assumption).

---

> > > > > > > ### Author Response · Authors · 2018-11-25
> > > > > > > **Further clarification on ERC**
> > > > > > >
> > > > > > > As we mentioned in the previous discussion, there is no conflict here. It is true that the empirical Rademacher based generalization bound implicitly assume that the training dataset is drawn from an underlying distribution that is same for the testing. But we want to remark that the quantities (especially B^Jac) in the upper bound of R_m only explicitly depends on the training dataset. More specifically, this comes from the fact that the empirical Rademacher complexity only depends on the training set directly. In other words, if the underlying distribution has good properties (e.g., data are reasonably distributed or separated), then the training set also inherit such good properties, and vice versa.
> > > > > > >
> > > > > > > Thanks for pointing out the notation. We have added a superscript on J to indicate the dependence on input. We updated the statement slightly by removing the uniform bound of the norm of Jacobian from the model class definition and define the corresponding quantity directly in the statement of the theorem. We hope this is more transparent from the reader’s point of view.
> > > > > > >
> > > > > > > The training set dependence in [1] also comes from the Dudley integral for bounding the empirical Rademacher complexity, where their margin depends on the training set. The dependence of the network compression on the training set only leads to log m factor in the number of compressed parameters due to the RIP type of properties the compressed network. The reason they analyze the compressed network is that they can reduce the number of parameters (that induces the generalization bound) to a quantity depending on the resilience parameters
> > > > > > >
> > > > > > > [1] Arora et al. Stronger generalization bounds for deep nets via a compression approach.

---

> > > > > > > > ### Public Comment · (anonymous) · 2018-11-26
> > > > > > > > **My concerns still remain**
> > > > > > > >
> > > > > > > > Dear authors,
> > > > > > > >
> > > > > > > > Thank you for considering my comments seriously and updating your paper. I'm afraid my concerns still remain.
> > > > > > > >
> > > > > > > >  1. About your claim "but as we have discussed [here on Openreview], this quantity empirically has a significantly weaker dependence than a quantity exponential on depth (even weaker than a low degree polynomial on depth)." My concern is that the paper lacks any experiment that justifies this claim. The only plot given shows the value of Bjac vs the spectral norm product for a particular D. A plot of the log of these values for depth is necessary to justify the paper's claim that Bjac is a constant w.r.t depth.
> > > > > > > >
> > > > > > > >  2. It seems like there's a misunderstanding about my point on ERC.
> > > > > > > >
> > > > > > > >   >>>  It is true that the empirical Rademacher based generalization bound implicitly assume that the training dataset is drawn from an underlying distribution that is same for the testing.
> > > > > > > >
> > > > > > > > In my discussion, I was not talking about differences in the train-time/test-time distributions at all.
> > > > > > > >
> > > > > > > > First, I think it'll be good to clarify the meaning of B_jac. To help me clarify the meaning, let's say X_m and X_m' are two different independent draws of the training datasets and let W the weights learned on X_m. Which of the following does B_jac bound?
> > > > > > > >
> > > > > > > > A. The spectral norm of the Jacobian of the weights W on X_m?
> > > > > > > > B. The spectral norm of the Jacobian of the weights W on X_m'?
> > > > > > > >
> > > > > > > > If your answer is both A and B then this is a really strong assumption that, as I said before, the paper should be clear about (the results as such will still hold). This is because if you assume B, you're effectively assuming that the network that is trained on a particular dataset, somehow has a specific kind of Jacobian on unseen data.
> > > > > > > >
> > > > > > > > If you answer is only A, then I believe there's a subtle bug in your proof. Perhaps it'll help if I can point out where I think the bug comes from. First of all in Lemma 2, you can have B_jac in the upper bound only if x belongs to the training dataset on which W was learned from (because you only assumed A!).  Next, in equation 13, L_w is an upper bound for all w and \tilde{w} in your hypothesis class, including weights that do not correspond to what you learned from the training dataset X_m. Therefore below Equation 15, it is incorrect to apply Lemma 2 (which specifically applies only when w is the weights learned on X_m).
> > > > > > > >
> > > > > > > > I'll be happy to further explain this if this is not clear.

---

> > > > > > > > > ### Author Response · Authors · 2018-11-26
> > > > > > > > > **please read the new revision**
> > > > > > > > >
> > > > > > > > > As we mentioned, we have updated the statement by removing the uniform bound of the norm of Jacobian from the model class definition in Eq. (5), i.e., we do not assume the uniform boundedness of Jacobian here.
> > > > > > > > >
> > > > > > > > > Instead, we define the corresponding norm of Jacobian (only depend on training data) directly in the statement of the theorem only. Please refer the new revision in terms of the Jacobian.
> > > > > > > > >
> > > > > > > > > Regarding the experiment, we will update it soon since it takes time.

---

> > > > > > > > > > ### Public Comment · (anonymous) · 2018-11-26
> > > > > > > > > > **Rephrased question B**
> > > > > > > > > >
> > > > > > > > > > Dear authors, I did go through the theorem statement. I'm sorry I wasn't clear, and I think I should rephrase B. I'm not sure I understand the meaning of B_jac. I understand it depends only on the training dataset. But what weights does it depend on?
> > > > > > > > > >
> > > > > > > > > > Which of the following does B_jac bound? Let W be the weights learned on a dataset X_m and W' be some other arbitrary weights. (I'm rephrasing  B here) Does B_jac bound:
> > > > > > > > > >
> > > > > > > > > > A. The spectral norm of the Jacobian of the weights W on X_m?
> > > > > > > > > > B'. The spectral norm of the Jacobian of any arbitrary weight W' on X_m?
> > > > > > > > > >
> > > > > > > > > > If you're assuming B', then it is strong and needs to be made clear as it's a norm bound on arbitrary weights that weren't learned on X_m.
> > > > > > > > > >
> > > > > > > > > > But without assuming B', I don't see how C_net can be written in terms of B_jac. I described the precise places where I think the analysis will be buggy in my last comment.
> > > > > > > > > >
> > > > > > > > > > Essentially, in either case, I have a concern and would really appreciate a clarification.
> > > > > > > > > >
> > > > > > > > > > About the experiments, great. And I totally understand it takes time. I hope you appreciate why I think it's necessary to substantiate the claim in the paper.

---

> > > > > > > > > > > ### Author Response · Authors · 2018-11-26
> > > > > > > > > > > **further update**
> > > > > > > > > > >
> > > > > > > > > > > Let me try to explain in another way. Our updated generalization bound is a data driven type of result, where the quantities (especially B^jac) in the bound do depend on the training data rather than taking the worst case. This is analogous to the result in Arora 18 that is also data driven, where their resilience parameters play similar roles to B^jac in terms of data dependence. In specific, if one considers the worst case, then the cushion parameter mu can be arbitrarily close to zero when one chooses a non-zero weight matrix (or Jacobian) almost orthogonal to some input, or the activation contraction parameter can be infinite is one chooses an input that has no active output in any layer. These can lead to an infinite generalization bound in their Theorem 4.1. As we have discussed, this data driven nature is the key to improving the generalization bound of DNNs, which avoids the worst-case scenarios.
> > > > > > > > > > >
> > > > > > > > > > > We also updated a contemporary experiment result for the dependence of B^Jac and product of norms on depth in a new revision for reference.

---

> > > > > > ### Public Comment · (anonymous) · 2018-11-24
> > > > > > **Clarification**
> > > > > >
> > > > > > At this point, I feel I should clarify that my intention is not to invalidate any of the results in the paper. I would only like to make sure that
> > > > > >
> > > > > > i) the authors are aware of the assumptions that I think have been implicitly made
> > > > > > ii) the paper is transparent about these assumptions throughout its text since these assumptions are strong (and this will be important for readers to compare and contrast the limitations/benefits of this work with others works),
> > > > > > iii) the paper justifies its claim about the precise dependence on depth
> > > > > > iv) or alternatively, I'm happy to be convinced by the authors that my understanding is wrong.
> > > > > >
> > > > > > Again, I appreciate the authors for having a discussion with me.

---

> > > > > > > ### Author Response · Authors · 2018-11-25
> > > > > > > **great discussion**
> > > > > > >
> > > > > > > Thanks for the clarification. It was great to discuss the details. It is true that the empirical Rademacher based generalization bound implicitly assume that the training dataset is drawn from an underlying distribution that is same for the testing. But we want to remark that there is no conflict with our previous discussion that the quantities (especially B^Jac) in the upper bound of R_m only explicitly depends on the training dataset. More specifically, this comes from the fact that the empirical Rademacher complexity only depends on the training set directly. In other words, if the underlying distribution has good properties (e.g., data are reasonably distributed or separated), then the training set also inherit such good properties, and vice versa.
> > > > > > >
> > > > > > > Moreover, we updated the statement by removing the uniform bound of the norm of Jacobian from the model class definition, i.e., we do not assume the uniform boundedness of Jacobian here. Instead, we define the corresponding norm of Jacobian (only depend on training data) directly in the statement of the theorem and we have empirically shown that this quantity is significantly smaller than the product of norms of weights. We hope this is more transparent from the reader’s point of view. In other words, we only assume the bounded spectral norm of weight. As we discussed earlier, the fact that Jacobian depends on both weights and inputs of training is an important reason that avoids us from meeting the worst-case bound as the product of norm. Therefore, we are not assuming anything about the Jacobian implicitly, but rather involve a quantity that depends on training data. From our point of view, a tighter generalization bound may not avoid such a quantity (depending on training data) since the global generalization bound that is independent of data may end up at a vacuous worst-case result.
> > > > > > >
> > > > > > > In addition, we are explicit on the dependence on depth in our result. I understand that you may worry about the dependence of the norm of Jacobian on the depth, as in some existing results [1,2] (where the product of norm in the log is much stronger than Jacobian and the output may also depend on depth), but as we have discussed, this quantity empirically has a significantly weaker dependence than a quantity exponential on depth (even weaker than a low degree polynomial on depth). The theoretical bound of this quantity is beyond the scope of this paper, which is an interesting future effort for us.
> > > > > > >
> > > > > > > I hope the discussion above clarifies your questions.
> > > > > > >
> > > > > > > [1] Arora et al. Stronger generalization bounds for deep nets via a compression approach.
> > > > > > > [2] Golowich et al. Size-Independent Sample Complexity of Neural Networks

---

> > ### Public Comment · (anonymous) · 2018-11-26
> > **My concerns about the data-driven approach (contd)**
> >
> > Thanks for considering my request and adding Figure 4. This figure is quite important to support your claims about explicit depth dependence in the main paper. I appreciate your update.
> >
> > Thanks for re-explaining your data-driven approach. I totally understand the need to consider a data-driven approach and I also understand that B_jac only involves the training data. What I'm looking for is an answer to my question about what weights are involved in the definition of B_jac and which of A and B' is assumed under my comment titled "Rephrased question B". You describe Bjac in Section 3, but the formulation currently is ambiguous. The notation Bjac is defined in terms of some W.  You later use this notation in theorem 1; what is the W you've implicitly used here in the definition of Bjac?  Is it the W learned on X_m?
> >
> > Just to help clarify what I mean there, Arora et al., assume bounded layer cushion etc., for a particular training dataset X_m and **a particular set of weights learned on that particular dataset X_m**. They do not assume layer cushion to be guaranteed to be nice on the training data X_m for other weights that the algorithm may learn on other draws of the datasets. When you say your assumptions are analogous to this, do you mean only assume A and not B' (see "Rephrased question B" for what I mean by A and B')?
> >
> > If the answer is yes, then I see a flaw in the application of Lemma 2 below Equation 15. I've described what I think is a flaw in "My concerns still remain" and would appreciate your response to that.
> >
> > I believe your analysis would work only if you assume that the Jacobian norm bound holds for X_m across many different W's, in which case, I'd recommend making this strong assumption apparent in the discussion.

---

> > > ### Author Response · Authors · 2018-11-27
> > > **further clarification**
> > >
> > > We have made it clear in our new revision that B^Jac is an upper bound of the Jacobian across the parameter space for a particular draw of training set X_m. This is also clear enough from the notation of the theorem and the analysis of the theorem. So B’ holds in your question. We have addressed discussion in the remark after the theorem regarding the bound on Jacobian.

---

> > > > ### Public Comment · (anonymous) · 2018-11-27
> > > > **Thanks for the clarification and for updating the paper**
> > > >
> > > > I find the theoretical analysis now consistent with the stated definitions/assumptions.
> > > >
> > > > I'd like to express my respect to the authors for taking the time, interest and effort to respond to all my questions throughout this long discussion. Thanks for the paper!

---

> > > > > ### Author Response · Authors · 2018-11-27
> > > > > **Thanks for your insightful comments**
> > > > >
> > > > > We really appreciate your very insightful suggestions for improving the paper. We have gained a much better understanding from your comments. : )

---

### Author Response · Authors · 2018-11-26
**Summary of updates in revision**

We remark a few important updates in the revision.

1. We updated some part of the analysis and the statement of the main theorem. Specifically, to bound the difference of two networks with respect to the difference of weight parameters, we use the norm of Jacobian rather than the product of norms of the weight matrices. Correspondingly, the log factor of the ERC bound depends on the spectral norm of Jacobian (rather than the product of norms) divided by the largest output of the loss function. The benefit of such an update is that the norm of Jacobian is significantly smaller than the product of norms of weight matrices in practice (the worst case that the norm of Jacobian close to the product of norms never happens in practice). To compare their difference, we provided an extended numerical result on the empirical distributions of the norm of Jacobian and the product of spectral norms in Appendix A.2. We observe that the values of the norm of Jacobian are orders smaller than the values product of norms. Our preliminary experiments also show that unlike the product of norms, the norm of Jacobian increases at a significantly slower rate than exponential on depth (even slower than some low degree of polynomial on depth, shown in Appendix A.3) when the depth increases. Thus, we can regard the log factor as a constant (almost) independent of depth. Moreover, the quantities in the log factor only depend on the training dataset, which is due to the fact that the ERC only depends on the training dataset. There is no conflict here with the implicit condition from the ERC based generalization bound that the training set is drawn an underlying distribution that is identical for testing. We have added Remark 1 for the discussion above.

2. For a more comprehensive numerical comparison, we have added an extended numerical result in Appendix A.1 for the bounded loss case to compare with [1,2]. We observe that our result is tighter than [1,2]. Moreover, the generalization bound in the bounded loss case in Figure 2 can be significantly smaller than the norm based result in Figure 1.

[1] Arora et al. Stronger generalization bounds for deep nets via a compression approach.
[2] Zhou and Feng. Understanding generalization and optimization performance of deep cnns.

---

### Meta-Review · Area_Chair1 · 2018-12-17
**Concerning forum**

**Confidence:** 3
**Recommendation:** Reject

**Metareview:**

I'm quite concerned by the conversation with Anonymous, entitled "Why is the dependence...". My issues concern the empirical Rademacher complexity (ERC) and in particular the choice of the loss class for which the ERC is being computed. This  class is obviously data dependent, but the Reviewers concerns centers on the nature of its data dependence. It is not valid to define the classes by the Jacobian's norm on the input data, as this _structure_ over the space of classes is data dependent, which is not kosher. The reviewer was gently pushing the authors towards a very strong assumption... i'm guessing that the jacobian norm over all data sets was bounded by a particular constant. This seems like a whopping assumption. The fact that I can so easily read this concern off of the reviewer's comments and the authors seem to not be able to understand what the reviewer is getting at, concerns me.

Besides this concern, it seems that this paper has undergone a rather significant revision. I'm not convinced the new version has been properly reviewed. For a theory paper, I'm concerned about letting work through that's not properly vetted, and I'm really not certain this has been. I suggest the authors consider sending it to COLT.